# Proteome and Metabolome Alterations in Radish (*Raphanus sativus* L.) Seedlings Induced by Inoculation with *Agrobacterium tumefaciens*

**DOI:** 10.3390/biom15020290

**Published:** 2025-02-14

**Authors:** Nadezhda Frolova, Daria Gorbach, Christian Ihling, Tatiana Bilova, Anastasia Orlova, Elena Lukasheva, Ksenia Fedoseeva, Irina Dodueva, Lyudmila A. Lutova, Andrej Frolov

**Affiliations:** 1Laboratory of Analytical Biochemistry and Biotechnology, K.A. Timiryazev Institute of Plant Physiology Russian Academy of Science, 127276 Moscow, Russia; frolovanadja@yandex.ru (N.F.); daria.gorba4@yandex.ru (D.G.); bilova.tatiana@gmail.com (T.B.); lanas_95@mail.ru (A.O.); 2Institute of Pharmacy, Martin-Luther University Halle-Wittenberg, 06099 Halle (Saale), Germany; christian.ihling@pharmazie.uni-halle.de; 3Department of Plant Physiology and Biochemistry, St. Petersburg State University, 199034 St. Petersburg, Russia; 4Department of Biochemistry, St. Petersburg State University, 199034 St. Petersburg, Russia; elena_lukasheva@mail.ru; 5Resource Center “Molecular and Cell Technologies”, St. Petersburg State University, 199034 St. Petersburg, Russia; fedoseeva.ksenia@gmail.com; 6Department of Genetics and Biotechnology, St. Petersburg State University, 199034 St. Petersburg, Russia; wildtype@ya.ru (I.D.); la.lutova@gmail.com (L.A.L.)

**Keywords:** agrobacterial infection, *Agrobacterium tumefaciens*, crown gall tumor, metabolic shifts, *Raphanus sativus* L., shotgun proteomics

## Abstract

Infection of higher plants with agrobacteria (*Agrobacterium tumefaciens*) represents one of the most comprehensively characterized examples of plant–microbial interactions. Incorporation of the bacterial transfer DNA (T-DNA) in the plant genome results in highly efficient expression of the bacterial auxin, cytokinin and opine biosynthesis genes, as well as the host genes of hormone-mediated signaling. These transcriptional events trigger enhanced proliferation of plant cells and formation of crown gall tumors. Because of this, infection of plant tissues with *A. tumefaciens* provides a convenient model to address the dynamics of cell metabolism accompanying plant development. To date, both early and late plant responses to agrobacterial infection are well-characterized at the level of the transcriptome, whereas only little information on the accompanying changes in plant metabolism is available. Therefore, here we employ an integrated proteomics and metabolomics approach to address the metabolic shifts and molecular events accompanying plant responses to inoculation with the *A. tumefaciens* culture. Based on the acquired proteomics dataset complemented with the results of the metabolite profiling experiment, we succeeded in characterizing the metabolic shifts associated with agrobacterial infection. The observed dynamics of the seedling proteome and metabolome clearly indicated rearrangement of the energy metabolism on the 10th day after inoculation (d.a.i.). Specifically, redirection of the energy metabolism from the oxidative to the anaerobic pathway was observed. This might be a part of the plant’s adaptation response to tumor-induced hypoxic stress, which most likely involved activation of sugar signaling.

## 1. Introduction

Infection of higher plants with the bacteria *Agrobacterium tumefaciens* (a causative agent of crown gall disease) represents one of the most comprehensively characterized examples of plant–microbial interactions [1]. Agrobacterial colonisation strategy relies on the integration of its T-DNA (a portion of the virulent plasmid) into the plant genome [2]. In nature, expression of agrobacterial *Vir* genes (which are directly involved in the regulation of the T-DNA) is induced by soluble phenols and sugars released from the plant tissues upon wounding [3]. The transfer of T-DNA and its integration into the plant genome trigger expression of agrobacterial genes encoding enzymes involved in of auxin, cytokinin (oncogenes *iaaH*, *iaaM*, and *ipt*) and opine biosynthesis [4,5] along with several host genes of hormone metabolism and signaling [2,6]. Therefore, the levels of cytokinins and auxins are increased in the infected tissues along with changes in dynamics of ethylene, abscisic acid and jasmonates [7]. This results in enhanced proliferation of the plant parenchyma cells and formation of the crown gall tumor [8]. These events are accompanied by enhanced expression of the genes which are constitutively involved in the development of meristems [9].

In this context, the interaction of plants with *A. tumefaciens* provides a convenient model to study the alterations in plant metabolism and meristem function that accompany plant development. Indeed, as tumor formation is underpinned by the loss of systemic control over cell proliferation, a comprehensive study of these aspects could provide insights into the mechanisms behind this control [10]. This knowledge is of principal importance, as the molecular mechanisms underlying tumor formation in plants are fundamentally different from those known in animals and occur much less frequently in nature [11].

Radish (*Raphanus sativus* L.) is a widely distributed root vegetable crop, which is commonly used as a model plant in growth and developmental research with a special emphasis on morphogenetic processes related to cell division, differentiation and expansion, as well as underlying signal transduction and metabolic pathways [9,12,13,14]. Post-genomic research methods (commonly referred to as “omics techniques”) represent a powerful toolbox in plant developmental biology and proved to be useful for studying *A. tumefaciens*-induced tumors in radish [15]. Thus, the proteomics data might provide insights into the functional relationships between changes in the genome and transcriptome and the characteristic morphological and biochemical alterations in phenotype [16].

Recently, a combination of transcriptomics and proteomics was employed to uncover the fine molecular mechanisms underlying growth and development in radish (*Raphanus sativus* L.). To achieve this, the transcriptome of the growing radish taproot was analyzed by RNA-seq in parallel to a comprehensive proteomics survey accomplished by iTRAQ [17]. The integration of two datasets gave access to the candidate functional proteins involved in taproot thickening in radish, which mostly represented hormone-related signal transduction, starch/sucrose metabolism, and specifically included regulators of cell division and expansion, including CDC5, EXPB1, and XTH24 [17].

Despite the impressive analytical power of modern proteomics, this methodology has only minimally been applied to the study of *A. tumefaciens*-induced tumors so far [17,18,19]. Indeed, highly-efficient liquid chromatography (LC)-based shotgun proteomics is still to be employed for the study of plant responses to *A. tumefaciens*. Because of this, the related changes in radish proteome are still insufficiently characterized. Therefore, here we report, to the best of our knowledge, the first comprehensive study of radish proteome dynamics in response to agrobacterial infection with a focus on late (i.e., those occurring after the first week post-inoculation) alterations in proteome associated with tumor development and accompanying metabolic and physiological changes.

## 2. Materials and Methods

### 2.1. Reagents

Unless stated otherwise, materials were obtained from the following manufacturers. AMRESCO LLC (Fountain Parkway Solon, OH, USA): ammonium persulfate (ACS grade), *N*,*N*’-methylene-bis-acrylamide (ultra-pure grade), potassium chloride (reagent grade), phenylmethylsulfonyl fluoride (PMSF, high purity grade); Bioanalytical Technologies 3M Company (St. Paul, MN, USA): Empore™ solid phase octadecyl extraction discs; Bio-Rad Laboratories (Moscow, Russia): Purezol reagent for RNA extraction, Quick start Bradford 1xDye Reagent; Carl Roth GmbH & Co (Karlsruhe, Germany): tris-(2-carboxyethyl)-phosphine hydrochloride (TCEP, ≥98%); Evrogen (St. Petersburg, Russia): primers for qRT-PCR; GE Healthcare (Piscataway, NJ, USA) : Electrophoresis reagent kit 2.D Quant; Helicon (Moscow, Russia): glycine (biotechnology grade), tris (hydroxymethyl) aminomethane (tris, ultra-pure grade), sodium dodecyl sulfate (SDS) (>99%), acrylamide (2K Standard Grade, AppliChem, Russia), sucrose (USP-NF Grade); PanReac AppliChem (Darmstadt, Germany): glycerol (ACS grade), acetonitrile (HPLC grade); Reachem (Moscow, Russia): hydrochloric acid (p.a.), isopropanol (reagent grade); SERVA Electrophoresis GmbH (Heidelberg, Germany): Coomassie Brilliant Blue G-250, 2-mercaptoethanol (research grade), trypsin NB (sequencing grade, modified from porcine pancreas); Syntol (Moscow, Russia): Eva Green intercalating dye; Thermo Fisher Scientific (Waltham, MA, USA): RapidOut DNA Removal Kit, Revert Aid Reverse Transcriptase kit, Pierce^TM^ Unstained Protein Molecular Weight Marker #26610 (14.4–116.0 kDa), PageRuller^TM^ Plus Prestained Protein Ladder #26620 (10–250 kDa); Vekton (Saint-Petersburg, Russia): conc. HCl (puriss). All other chemicals were purchased from Sigma-Aldrich Rus LLC (Moscow, Russia): ethanol (200 proof, for molecular biology), hydrogen peroxide (puriss. p.a.), Luria-Bertani medium (LB) (grade for molecular biology), isopropanol (grade for molecular biology), chloroform (for HPLC), agar (extra pure). Water was purified in-house (resistance 5–15 mΩ/cm) on an Elix 3 UV water conditioning and purification system (Millipore, Moscow, Russia).

### 2.2. Plant Material

Inbred line 18 from the genetic collection of radishes (*Raphanus sativus* L.) obtained from St. Petersburg State University was used in this study [20]. This line is approximately the 40th inbred generation, originating from *Saxa* cultivar (European group of radish varieties).

### 2.3. Plant Growing

Radish seeds (at least 100) were surface-sterilized with a 1:1 *v*/*v* mixture of 90% (*v/v*) ethanol and 30% (*v*/*v*) hydrogen peroxide for seven minutes, germinated in Petri dishes on solid Murashige–Skoog (MS) medium [21] with 8 g/L agar and grown for seven days at 21 °C under a 16:8 h day/night regimen before the seedlings were inoculated with *A. tumefaciens.*

### 2.4. Inoculation of Radish Seedlings with Agrobacterium tumefaciens

The bacteria (*A. tumefaciens* strain C58) were grown in liquid LB medium for two days with constant aeration on a rotary shaker at 28 °C to obtain optical density (OD) 1–1.5 at 600 nm. Inoculation of the seven-day-old aseptic radish seedlings (all germinated seedlings were used) with agrobacterial culture was accomplished by application of 2–3 μL of *A. tumefaciens* suspension on hypocotyl incisions freshly made with a sterile blade. Sterile LB medium was used for mock-inoculation of seedlings (control treatments). On the fifth day after inoculation, the seedlings were transplanted onto a medium containing cefotaxime (400 mg/L) to eliminate the remaining bacteria.

### 2.5. Methods of Protein Isolation and Tryptic Digestion

The frozen seedlings were ground in liquid nitrogen using a Mixer Mill MM 400 ball mill with Ø 3 mm stainless steel balls (Retsch, Haan, Germany) at a vibration frequency of 30 Hz twice for one min. The total protein fraction was isolated from the frozen material (approximately 250 mg) by the phenol extraction method, as described by Mamontova et al. [22] (for details see Appendix A). The protein isolates were dried in a fume hood for 30 min and re-constituted in 100 µL of shotgun buffer (7 mol/L urea and 2 mol/L thiourea in 50 mmol/L Tris-HCl buffer, pH 7.2), containing 0.15% (*w*/*v*) Anionic Acid Labile Surfactant II (AALS II). Determination of protein concentrations relied on the protein-catalyzed reduction of cupric (Cu^2+^) ions to cuprous (Cu^+^) ions with the 2-D Quant kit (GE Healthcare Bio-Sciences AB, Sweden) as described by Mamontova et al. [23]. The results of protein determination were cross-validated with SDS-PAGE, as described before [24].

Tryptic digestion relied on the protocol of Matamoros and co-workers [25] with minor modifications. In detail, aliquots of protein (20 µg) were supplemented with 10 µL of TCEP (50 mmol/L) in the AALS-free shotgun buffer and diluted to obtain a total volume of 100 µL. After a 30 min incubation at 37 °C under continuous shaking (250 rpm) and cooling the samples to room temperature (RT), 11 µL of 0.1 mol/L iodoacetamide in shotgun buffer was added, and the mixtures were incubated for 60 min at 4 °C in darkness. Afterwards, 875 µL of 50 mmol/L ammonium bicarbonate were supplemented to each sample, and trypsin (0.5 g/L in the same solution) was sequentially added twice at enzyme–protein ratios of 1:20 and 1:50. Proteolysis was accomplished at 37 °C under continuous shaking (250 rpm) for periods of 5 and 12 h. The completeness of tryptic digestion was confirmed by SDS-PAGE (as an orthogonal analytical method of cross-validation) [24]. AALS was destroyed by addition of 111 µL of 10% (*v*/*v*) trifluoroacetic acid (TFA, final concentration 1% *v*/*v*) and incubation for 20 min at 37 °C under continuous shaking (450 rpm). After this, the digests were desalted by solid phase extraction (SPE) using in-house prepared stage-tips, i.e., polypropylene pipette tips (200 µL) filled with six layers of C18 reversed phase material (Empower™ SPE discs) as described by Mamontova et al. [23] using the elution protocol of Spiller et al. [26]. The resulting eluates were freeze-dried overnight under reduced pressure in a CentriVap Vacuum Concentrator (Labconco, Kansas City, MO, USA) and stored at −20 °C before analysis.

### 2.6. nanoHPLC-LIT-Orbitrap-MS/MS

For LC-MS analysis, the whole sample set was randomized and standardized to quality controls (QCs, injected after each six samples)—aliquots of a pool obtained by mixing 10 µL of each tryptic digest. Individual digests (500 ng, 10 µL) dissolved in 3% (*v*/*v*) acetonitrile in 0.1% (*v*/*v*) aq. TFA were loaded onto an Acclaim PepMap 100 C_18_ trap column (300 µm × 5 mm, 3 µm particle size) during 15 min at a flow rate of 30 µL/min. The proteolytic peptides were separated at a flow rate of 300 nL/min on an Acclaim PepMap 100 C_18_ column (75 µm × 250 mm, particle size 2 µm) using an Ultimate 3000 RSLC nano-HPLC system coupled on-line to a hybrid LTQ Orbitrap XL mass spectrometer via a nano-ESI source equipped with a 30 µm i.d., 40 mm long steel emitter (all Thermo Fisher Scientific, Bremen, Germany). The eluents A and B were 0.1% (*v*/*v*) aq. FA and 0.08% (*v*/*v*) FA in acetonitrile, respectively. The peptides were eluted with linear gradients ramping from 1 to 35% eluent B over 90 min followed by 35 to 85% eluent B over 5 min. The column was washed for 5 min, and re-equilibrated at 1% eluent B for 10 min. The nanoLC-MS analysis relied on data-dependent acquisition (DDA) experiments performed in the positive ion mode, comprising a survey Orbitrap-MS scan and MS/MS scans for the most abundant signals in the following 5 s (at certain t_R_) with charge states ranging from 2 to 6. The mass spectrometer settings and DDA parameters are summarized in the Appendix A.

### 2.7. Protein Annotation and Label-Free Quantification

Identification of peptides and annotation of proteins relied on a search with the SEQUEST engine (run under Proteome Discoverer 2.2 software, Thermo Fisher Scientific, Bremen, Germany) against the sequence databases *Raphanus sativus* database (NCBI, UniProt, and 2015 *R. sativus* genome assembly [14] uploaded on 18 June 2019) using the settings specified in Appendix A. Venn diagrams were constructed for identified peptides, proteins and non-redundant proteins (protein groups) using the environment of computational programming R software environment for statistical computing and graphics Version 4.0 (https://www.r-project.org/) and *VennDiagram* package Version 1.7.3 (https://cran.r-project.org/web/packages/VennDiagram/, accessed on 12 April 2022).

Label-free quantification relied on the Progenesis QI software Version 4.2 (Waters GmbH, Eschborn, Germany). After peak peaking and alignment, appropriate spectral and peptide filters (charge 2–6, ANOVA/*t*-test at *p* ≤ 0.05, q ≤ 0.05, rank > 3, coefficient of variation CV ≤ 60%, hide no MS/MS data) were applied, and selected MS/MS spectra were searched against the combined *Raphanus sativus* database (NCBI, Uniprot and 2015 *R. sativus* genome assembly uploaded on 18 June 2019) with methionine oxidation as the only dynamic modification. All retrieved protein sequences were fused to obtain one fasta file. Thereby, the sequence redundancy was eliminated by database clustering using CD-HIT algorithm with sequence identity cut-off set to 1 (http://cd-hit.org/). Afterwards, the MGF file was re-loaded to the Progenesis QI software, and the filtered list of sequences identified as differentially expressed proteins was exported. QIP filters: ANOVA/*t*-test *p* ≤ 0.05, q ≤ 0.05, fold changes (FC) ≥ 1.5 for at least one group comparison. The proteins meeting the filtering criteria were exported for statistical interpretation in R studio Version: 2024.12.0+467 (RStudio Desktop-Posit). Data pre-processing included logarithmical (log_2_a) transformation and quantile normalization of raw analyte abundances imported from the Progenesis QI software.

Functional annotation relied on Mercator v3.6 (https://plabipd.de/portal/mercator-sequence-annotation, accessed on 15 November 2013) with default parameters, whereby, to all proteins one or several of 35 functional terms (bins) were assigned. Prediction of cellular localization relied on the BUSCA (Bologna Unified Subcellular Component Annotator) tool (http://busca.biocomp.unibo.it/, accessed on September, 2022). Additional functional annotation was obtained from the KEGG (Kyoto Encyclopedia of Genes and Genomes) database using R package KEGGREST Version 1.46.0 (Bioconductor—KEGGREST, https://bioconductor.org/packages/release/bioc/html/KEGGREST.html, accessed on February, 2012) and BLAST Koala Version 2.2 (BlastKOALA, https://www.kegg.jp/blastkoala/, accessed on May, 2019). To increase the number of annotated protein groups, all sequences within a protein group were used for prediction of both localization and function. All annotations and predictions were manually verified according to their accession numbers using free databases: UniprotKB (UniProt), nextprot database (neXtProt Search), BRENDA Enzyme Database—BRENDA (BRENDA professional, https://brenda-enzymes.com/, accessed on December, 2024).

### 2.8. Analysis of Primary Metabolome

Primary thermally stable and thermally labile metabolites were analyzed by gas chromatography-quadrupole mass spectrometry with EI ionization (GC-EI-Q-MS) and ion-pair reversed-phase high-performance liquid chromatography coupled online to triple quadrupole tandem mass spectrometry (RP-IP-HPLC-QqQ-MS/MS) accomplished as multiple reaction monitoring (MRM) experiments, respectively, as described by Shumilina et al. [27]. The corresponding instrument-specific GC and LC parameters and MS settings are summarized in Appendix A, respectively.

Processing of the GC-MS data and relative quantification relied on the Automated Mass Spectral Deconvolution and Identification System, AMDIS v.2.66 (www.amdis.net, accessed on 10 March 2023) and MSDial (https://prime.psc.riken.jp/compms/msdial/, accessed on 10 March 2023). Identification of trimethylsilyl (TMS) and methyl oxime (MEOX)-TMS derivatives of the analytes relied on co-elution with authentic standards or/and spectral similarity search against available spectral libraries—NIST (National Institute of Standards and Technology, https://webbook.nist.gov/chemistry/, updated on January 2023), GMD (Golm Metabolome Database, http://gmd.mpimp-golm.mpg.de/, updated on 31 August 2021) and in-house spectral library (partially with Kovats retention time indices, calculated by the retention times of alkane standards). Annotation of unknowns relied on characteristic *m*/*z*, t_R_ and retention indices (RIs). Annotation of individual analytes in LC-MS chromatograms relied on co-elution with authentic standards in corresponding MRM scans, whereas relative quantitative analysis assumed integration of corresponding chromatographic peaks at specified transitions and t_R_s with PeakView 2.2 Software (PeakView Software Version 2.2.0.11391, AB SCIEX, 2014) and MultiQuant 3.0.2 Software (MultiQuant Software Version 3.022850.0, AB SCIEX, 2015) (Appendix A). After processing, the data matrices (including peak areas at specified retention time t_R_) obtained from the LC-MS and GC-MS analyses were combined and post-processed.

Data post-processing relied on the methods of multivariate and invariant statistics implemented in the MetaboAnalyst 6.0 on-line tool (MetaboAnalyst). Prior to the statistical analysis, the data were normalized to the dry weight of samples (DW normalization). For metabolites not detected in <20% of samples, missing value imputation was performed using the KNN (k nearest neighbor) algorithm (sample-wise).

## 3. Results

### 3.1. Induction of Crown Galls

On the 10th day after inoculation (d.a.i.) with *A. tumefaciens*, visible tumors could be distinguished on the hypocotyls of only some seedlings (Figure 1A). In contrast, callus-like crown gall tumors (Ø 1–5 mm) could be clearly observed near the most inoculation sites on the 22nd d.a.i. (Figure 1B). In control mock-inoculated seedlings, incision sites with sporadic necrosis regions could be rarely observed (Figure 1C).

### 3.2. Protein Isolation and Tryptic Digestion

The proteins were successfully isolated from the radish seedlings with yields of 0.81–6.96 mg/g FW (protein concentrations in the extracts were 1.45–13.57 mg/mL, Appendix A). These results were cross-validated by SDS-PAGE with Coomassie staining. Thereby, the sample load was calculated based on the results of the protein determination assay with 2D-Quant kit. The whole lane average intensities obtained with equal sample load (5 µg of protein per each lane) were 1.84 × 10^4^ ± 4.99 × 10^3^ (relative standard deviation RSD = 27.1%). However, as can be seen from Appendix A, the samples C22-2 and Ex22-3 demonstrated intensity at least two-fold higher than the average value. This might indicate mistakes in protein determination. This assumption was confirmed by an outlier test. Indeed, after removal of these two values the statistics noticeably improved: the average intensity was 1.70 × 10^4^ ± 2.39 × 10^3^, and RSD value was two times lower (14.1%). Based on our previous experience, this result can be considered acceptable. Therefore, all protein samples were digested by two sequential treatments with trypsin as described in the Materials and Methods section. The digestion was considered to be completed, as the RuBisCO large subunit band was not detectable anymore. This corresponded to a digestion efficiency better than 95% (assuming that the sensitivity of staining was not worth than 30 ng).

### 3.3. Annotation of Radish Proteins

A search for the acquired data-dependent acquisition (DDA) raw files (available in the PRIDE repository under the project accession number PXD060463 and project DOI 10.6019/PXD060463) within the radish sequence database resulted in MS/MS-based identification of 2642 and 4098 peptides (Appendix A), found in the groups harvested on the 10th and 22nd days after inoculation with agrobacterial culture, respectively. On the other hand, 2999 and 4056 peptides could be found in corresponding mock-treated control groups. These peptides could be assigned to 2609 and 3732 possible individual proteins in the seedlings harvested on the 10th and 22nd day after inoculation with agrobacteria, respectively (Appendix A), whereas 2792 and 3585 proteins could be annotated in corresponding mock treatments. Among the proteins annotated in the groups treated with agrobacterial culture for 10 and 22 days, 894 and 1281, respectively, were non-redundant (so-called protein groups), whereas 957 and 1253 non-redundant proteins were detected in the corresponding medium-treated groups (Figure 2, Appendix A). Among identified non-redundant proteins, 580 were found in all groups, whereas 459 were found only in agrobacterial treatment groups.

It is necessary to take into account, however, that DDA-based identification without matching of individual peptide signals across the whole dataset does not allow assessment of differential expression patterns and drawing of reliable conclusions about the specificity of individual proteins for agrobacterial treatment. To access these data, label-free quantification was accomplished.

### 3.4. Label-Free Quantification

Label-free quantitative analysis relied on Progenesis QI software and normalization by protein load based on determined concentrations (Appendix A). The success of normalization was verified by pooled quality controls (QCs) injected after each six samples (by integration of several randomly selected peptides in corresponding XICs). The analysis revealed, in total, 487 differentially expressed (in comparison to the corresponding mock treatments) proteins (111 and 376 on the 10th and 22nd day after the inoculation, respectively, Appendix A). On the 10th day, 25 polypeptides were up-regulated, among which the contents of glyceraldehyde-3-phospate dehydrogenase (296.4-fold) and aconitate hydratase (infinite) were the most strongly changed (Appendix A). Approximately three times more (83) proteins were down-regulated at this time point, with 60S ribosomal protein L22-2 being the most affected (six-fold change, Appendix A).

The number of significantly regulated proteins dramatically (approximately four-fold) increased till the 22nd day. Thereby, in contrast to the first time point, the numbers of up- and down-regulated species was comparable. Thus, 195 proteins were up-regulated (Appendix A), with glyceraldehyde-3-phospate dehydrogenase (50-fold) and aconitate hydratase (135-fold) remaining the most affected along with strong up-regulation of 3-oxoadipate CoA-transferase (30-fold). Among the 181 down-regulated species (Appendix A), non-phosphorylating glyceraldehyde 3-phosphate dehydrogenase (NPGAP-DH), TonB-dependent siderophore receptor and ferredoxin--NADP reductase were the most affected (15.4, 17.3 and 14.9-fold, respectively).

To assess the contribution of agrobacterial infection in dynamics of individual proteins in early plant ontogenesis, we compared the changes in the seedling proteome across the whole experiment time in both presence and absence of agrobacterial infection. For this, we addressed the time-related patterns of differential expression obtained in absence (mock inoculation with culture medium) and presence (inoculation with *A. tumefaciens* culture).

Comparison of the plants harvested on the 10th and 22nd d.a.i. revealed significant differences when both agrobacterial and mock treatments (374 and 122 differentially expressed proteins, respectively) were compared (Appendix A). Comparison of the plants inoculated with agrobacterial culture revealed 299 up-regulated proteins, the most affected of which were large subunit ribosomal protein L13Ae (15.7-fold) and fructose-bisphosphate aldolase (11.9-fold, Appendix A). Among the 75 down-regulated accessions, the most strongly changed were photosystem I reaction center subunit IV A, chloroplastic-like (7.8-fold), Short-chain dehydrogenases/reductases (SDR) family NAD(P)-dependent oxidoreductase (5.6-fold) and annexin D1 (7.3-fold, Appendix A).

The control (mock-treated) plants showed a different pattern of age-related differential protein expression. Thus, 102 proteins were up-regulated in control treatments on the 22nd day in comparison to the 10th, with *D*-3-phosphoglycerate dehydrogenase (52.3-fold) and chloroplastic glyceraldehyde-3-phosphate dehydrogenase GAPCP1 (9.9-fold) being the most affected (Appendix A). The pattern of down-regulated proteins in control incubations (20 entries, Appendix A) dominated with cytosolic glyceraldehyde-3-phosphate dehydrogenase (3.2-fold) and homeobox-leucine zipper protein HDG12, isoform X1 (3-fold).

### 3.5. Functional Annotation of Differentially Expressed Proteins

The proteins differentially expressed on the 10th and 22nd days after agrobacterial inoculation represented 27 and 32, respectively, of the 35 Mercator/MapMan functional groups (so-called bins, Figure 3) totally available for annotation.

The majority of the 25 proteins up-regulated on the 10th d.a.i. with *Agrobacterium tumefaciens* suspension in comparison to the culture medium (in total, 14 functional bins), were annotated to bins 26 (miscellaneous enzyme families), 21 (redox, dominated by peroxidases), 4 (glycolysis), 8 (TCA/organic acid transformation) and 16 (secondary metabolism, Figure 3 left, Appendix A).

The 83 proteins down-regulated on the 10th d.a.i. with *Agrobacterium tumefaciens* suspension in comparison to the culture medium were annotated to 22 bins (Figure 3 left, Appendix A) with functional classes related to bins 29 and 34 (protein metabolism and transport—15 and 11 accessions, respectively) being the most represented. Bin 29 mostly included structural proteins of 40S and 60S ribosomes, as well as initiation translation factors (Appendix A). Bin 34 transport was predominantly annotated to V-ATPases, with a minor contribution of the general substrate transporter and alanine dehydrogenase. Among transport proteins, the most affected was the general substrate transporter (sugar transporter ERD6-like 3) with a 4.1-fold increase. The further six proteins of amino acid biosynthesis (bin 13) and five glycolytic enzymes (bin 4) were down-regulated with glycine cleavage system P protein and NADP-dependent glyceraldehyde-3-phosphate dehydrogenase demonstrating the strongest decrease (2.9- and 5.0-fold, respectively) (Appendix A). The proteins involved in photosynthesis (bin 1; four proteins, up to 2.8-fold down-regulation), carbohydrate metabolism (bin 2 and 3; eight proteins, up to 3.9-fold), TCA/organic acid transformation (bin 8; three proteins, up to 2.6-fold), cell wall (bin 10; four proteins, up to 4.4-fold) and signal transduction (bin 30; 4 proteins; up to 3.3-fold) were less affected, although still well represented in the patterns of differential expression (Appendix A).

The proteins up-regulated on the 22nd d.a.i. with *Agrobacterium tumefaciens* suspension in comparison to the mock treatment (in total 26 bins) were strongly dominated by bin 29 (protein metabolism), which was represented with 91 accessions accounting for approximately half of all the up-regulated polypeptides (Figure 3 right, Appendix A). These included 64 structural components of 40S and 60S ribosomes, nine factors of translation initiation and elongation, eight chaperones and ten actors of protein degradation machinery. Among these four groups, chaperones were the most up-regulated (the most affected was the proteasome assembly chaperone 2 with a 13.1-fold change), whereas the other proteins, related to the proteasomal degradation machinery, appeared to be the least influenced. The most regulated among the latter group was the 26S proteasome non-ATPase regulatory subunit 13 homolog B, which was 2.5-fold up-regulated.

Much less numerous, but still well-represented, were the proteins involved in stress response (bin 20; eight proteins) and redox processes (bin 21; 13 proteins), as well as the members of the “miscellaneous enzyme families” group (bin 26; four proteins). Stress/redox-related proteins were strongly elevated—particularly peroxidases (up to 11.8-fold for the peroxidase 37-like) and oxidoreductases (up to 6.7-fold for NAD(P)-dependent oxidoreductase of the SDR family). The contribution of other functional groups—major carbohydrate metabolism, OPP cycle, TCA/organic acid transformation, mitochondrial electron transport/ATP synthesis proteins, cell wall, lipid metabolism, amino acid metabolism, secondary metabolism, DNA metabolism, signaling and cell organization, contributed much less to the overall pattern of up-regulated proteins (23% in total).

The late patterns of protein down-regulation (in comparison to the mock treatment—26 bins in total), acquired on the 22nd d.a.i., were more balanced and dominated by polypeptides involved in amino acid biosynthesis (bin 13), which accounted for in total 21 protein accessions demonstrating up to 9-fold agrobacteria-related abundance decrease, particularly for glutamine synthetase (Appendix A). Slightly less numerous, but nevertheless highly represented, were the proteins of the following functional groups: photosynthesis (bin 1; 13 proteins), redox (bin 21; 15 proteins), cell organization (bin 31; 10 proteins) and transport (bin 34; 11 proteins), while the remaining functional groups were represented by fewer proteins—from one to nine proteins per group. As far as the quantitative aspect is concerned, the representatives of bins 10 and 4 were the most strongly down-regulated (up to 15-fold for xyloglucan endotransglucosylase/hydrolase and up to 15.4-fold for NADP-dependent glyceraldehyde-3-phosphate dehydrogenase).

Analysis of the time domain in the protein dynamics revealed rich patterns of protein expression both in the presence and absence of agrobacterial infection (Appendix A). Thus, functional annotation revealed 30 groups of proteins (bins) differentially regulated on the 22nd d.a.i. relative to the earlier sampling point on the 10th d.a.i. (Appendix A). Thereby, 27 bins (299 proteins) and 18 bins (75 proteins) were up- and down-regulated specifically in infected seedlings, whereas 20 bins (102 proteins) and 13 bins (20 proteins) were up- and down-regulated in mock treatments.

On the 22nd d.a.i., the treatment with cultural medium (mock) resulted in essential up-regulation (in comparison to the 10th d.a.i.) of bin 29 (protein metabolism, 29 accessions, up to 8-fold for elongation factor 1-alpha, accession # RSG24111.t1) as well as, to less extent, of bin 4 (glycolysis, eight accessions, up to 9.9-fold for glyceraldehyde 3-phosphate dehydrogenase), bin 13 (amino acid metabolism, 14 accessions, up to 52.4-fold for *D*-3-phosphoglycerate dehydrogenase), and, finally, bins 20 (stress, eight accessions) and 21 (redox, seven accessions) with the most pronounced changes observed for the peroxidase 32 and haem peroxidase (4.3- and 3.5-fold, respectively), whereas the impact of the other groups was minor (Appendix A).

The constitutive down-regulation patterns on the 22nd d.a.i. (which were four times less rich) were dominated by bin 4 (glycolysis, three accessions, up to 3.2-fold for glyceraldehyde 3-phosphate dehydrogenase), and bin 29 (protein metabolism, eight accessions, up to 2.3-fold for GTP-binding domain-containing elongation factorRSG13225.t1) with only several (mostly one or two) proteins regulated in other functional groups (Appendix A).

Similarly to the controls, the treatment with agrobacterial suspension resulted 22 days later in essential up-regulation (in comparison to the 10th d.a.i.) of bin 29 (Appendix A), which was four times stronger in comparison to the former (133 accessions, up to 15.7-fold for ribosomal protein eukaryotic 60S subunit L13A). This group was represented by the polypeptides directly involved in the initiation and elongation of translation, formation of ribosomes, folding and protein degradation. Several functional groups contributed to the up-regulation patterns with 15–20 proteins: bin 4 (glycolysis, 15 accessions, up to 11.9-fold for fructose-bisphosphate aldolase), bin 13 (amino acid metabolism, 18 accessions, up to 3.6-fold for amidohydrolase), bin 20 (stress, 11 accessions, up to 3.2-fold for DEAD-box ATP-dependent RNA helicase 2-like), bin 21 (redox, 14 accessions, up to 10-fold for peroxidase 37-like), bin 23 (nucleotide metabolism, 11 accessions, up to 9.7-fold for mitochondrial dihydroorotate dehydrogenase) and bin 34 (transport, 15 accessions, up to 9.1-fold for sugar transporter ERD6-like 3 protein). The further 18 bins were represented with one to nine proteins with fold change range of 1.4-fold for cell division control protein 48 homolog A-like—9.3-fold for NADH-quinone oxidoreductase subunit F.

The profile of time-dependent down-regulation associated with agrobacterial infection (Appendix A) was dominated by bin 21 (redox, nine accessions, up to 5.6-fold for dehydrogenase/reductase SDR family member 7C) and bin 1 (photosynthesis, 10 accessions, up to 7.8-fold for chloroplastic-like photosystem I reaction center subunit IV A) with lower contribution of other groups.

### 3.6. Prediction of Cellular Localization of Differentially Expressed Proteins

To assess the spatial domain of the proteome changes associated with the agrobacterial infection, we addressed intracellular localization of differentially expressed proteins by means of predictive bioinformatics tools (Appendix A). The comparison of the inoculated plants with the corresponding mock treatments on the 10th and 22nd d.a.i. allowed assignment of characteristic localization patterns.

The changes occurring on the 10th d.a.i. were mostly localized to the cytoplasm/cytosol which accounted in total for about 35% of all proteins annotated as up- and down-regulated (Figure 4A,B). Analogously, in terms of the relative numbers of protein accessions, the impact of plasma membrane, chloroplasts, endomembrane system and nucleus was similar in both regulation groups, whereas the absolute numbers of down-regulated proteins in each set of localizations was approximately 3-fold higher in the down-regulated group (Appendix A). Interestingly, peroxisomal and proteasomal proteins demonstrated a clear up-regulation pattern at this time point (Figure 4A), whereas ribosomal and vacuolar proteins showed a clear down-regulation profile on the 10th day (Figure 4B). The same was the fact for apoplast, cell wall, cytoskeleton, Golgi apparatus and extracellular matrix (Figure 4B).

On the 22nd d.a.i., the patterns of differential expression changed dramatically in the localization aspect: although the cytosol and nuclear proteins still represented approximately one third and one sixth of all differentially expressed proteins, respectively, the contribution of other compartments changed essentially (Figure 4C,D, Appendix A). Thus, the polypeptides of ribosomes and ER were strongly up-regulated (26 and 6%, respectively, Figure 4C), whereas their presence in the down-regulation patterns was negligible (0 and 2%, respectively, Figure 4D). On the other hand, the proteins of cytoskeleton, plasma membrane, chloroplasts and mitochondria were strongly down-regulated (4, 7, 11 and 20%, respectively), that was two to four times higher in comparison to the corresponding up-regulation patterns (1, 3, 4 and 5%, respectively, Figure 4D). Similarly to the first time point, only the proteins of proteasomes and peroxisomes were up-regulated, whereas all differentially expressed reserve polypeptides (constituting lipid and protein droplets) and secretory proteins were down-regulated (Figure 4C,D).

At the next step, we compared age-dependent alterations (22 d.a.i. vs. 10 d.a.i.) in the proteome signatures of radish seedlings, observed in the presence and absence of agrobacterial infection (Appendix A). Thus, the mock-treated seedlings demonstrated a strong increase of cytosolic, mitochondrial, chloroplast and nuclear proteins (constituting 36, 13, 12 and 12% of the total up-regulated proteome, respectively) on the 22nd in comparison to the 10th day of the treatment with bacteria-free cultural medium (Appendix A). In contrast, the age-dependent down-regulation profiles induced by the mock treatment were strongly dominated, besides cytosolic, by ribosomal proteins (22%), whereas nuclear, mitochondrial and chloroplast polypeptides were less represented in this pattern (Appendix A). Interestingly, all differentially expressed apoplast proteins were down-regulated (7%) whereas ER proteins (8%) were only up-regulated (Appendix A).

Inoculation with agrobacterial culture dramatically altered age-related patterns of differential expression. Thus, in contrast to the mock treatment, agrobacterial infection resulted in pronounced up-regulation of ribosomal proteins, which was not observed when the bacteria were not supplemented to the inoculation solution (Appendix A). On the other hand, the contribution of chloroplast proteins in age-dependent up-regulation patterns was much less in comparison to the mock treatment (4 vs. 12%). The same was true for vacuolar and ER polypeptides (Appendix A).

The effect of the agrobacterial infection on the down-regulation patterns was not less pronounced. Thus, in contrast to the mock treatment, no effect on ribosomal proteins was observed, whereas the contribution of chloroplast and plasma membrane proteins was higher (15 vs. 8 and 9 vs. 4%, respectively, Appendix A). Remarkably, in contrast to the mock treatment, secretory proteins were time-dependently down-regulated. Moreover, multiple unique minor localization contributors (vesicles, apoplast, cell wall, endomembrane system and others) could be observed in the protein down-regulation pattern in the presence of agrobacterial infection.

### 3.7. Targeted Analysis of the Meristem-Related Proteins

Since meristems represent the most important conserved morphological module regulating plant growth, histogenesis and organogenesis [28], we addressed the dynamics of meristem-related proteins in more detail. For this, we employed a targeted strategy, which covered in total 37 peptides and proteins from *R. sativus*, involved in/related to meristem differentiation (Appendix A). The selection of the proteins relied on data from previous studies using inbred radish lines to analyze differential gene expression in the crown gall induced by *A. tumefaciens* [15] and also in the spontaneous radish tumors compared to lateral roots using the RNA-sequencing (RNA-seq) method. Then, the whole dataset was searched against a database constructed from the sequences of these proteins. This yielded identification of a total of 51, 72 and 62 peptides and proteins and non-redundant proteins (protein groups), respectively, involved in regulation of tissue growth and development in the context of meristem activity. Among this number, a total of 15 species was assigned as differentially expressed (Table 1). The correctness of their identification was verified by manual interpretation of the corresponding MS/MS spectra (Appendix A). Thereby, seven (poly)peptides were differentially expressed in the agrobacteria-treated plants in comparison to the controls at least at one time point, whereas the remaining eight demonstrated no differences with controls at both sampling times, but showed alterations in protein dynamics in the presence of *A. tumefaciens* infection.

The former group of seven accessions was represented by four clusters, which were formed according to the occurrence of individual proteins among differentially expressed paired comparisons (see explanation for Table 1): (i) the proteins up-regulated in the presence of agrobacterial infections only on the 10th d.a.i. (CLAVATA3/ESR-RELATED 22), (ii) those up-regulated both on the 10th and 22nd d.a.i. (CLAVATA3/ESR-RELATED 45), (iii) down-regulated only on the 10th d.a.i. (CLAVATA3/ESR-RELATED 27, CYCLIN D3;3 and GRAS family transcription factor NP_191622.1) and (iv) the species increasing their abundance on the 22nd d.a.i. (GRAS family transcription factor NP_195389.4 and CLAVATA3/ESR-RELATED 5).

The other eight accessions did not show any agrobacteria-related statistically significant abundance differences on the 10th and 22nd d.a.i., although demonstrated differential age-related dynamics in the presence and absence of inoculation with *A. tumefaciens*. This group was represented with CLAVATA3/ESR-RELATED peptides (CLE 21, CLE 19 and CLE 1), WUSCHEL-related homeobox peptides 5 and 14, homeobox protein knotted-1-like 3 and homeobox-leucine zipper protein ATHB-8 proteins, as well as cyclin D3. Among these proteins, the abundance of CLAVATA3/ESR-RELATED 21 and CLAVATA3/ESR-RELATED 1 slightly time-dependently increased, whereas the relative contents of WUSCHEL-related homeobox decreased in mock treatments. On the other hand, the abundance of WUSCHEL-related homeobox 5, WUSCHEL-related homeobox 14 and homeobox protein knotted-1-like 3 slightly time-dependently increased (Table 1).

### 3.8. Analysis of the Primary Metabolome

The analysis of primary metabolites relied on a combination of two methods: gas chromatography-quadrupole mass spectrometry with EI ionization (GC-EI-Q-MS) and ion-pair reversed-phase high-performance liquid chromatography coupled online with triple quadrupole tandem mass spectrometry (RP-IP-HPLC-QqQ-MS/MS). After processing, both datasets were merged into one matrix prior to statistical evaluation.

The untargeted GC-EI-Q-MS approach relied on aqueous methanolic extracts and could give access to thermally stable primary metabolites, i.e., to the part of the primary metabolome of radish calluses which did not undergo thermal degradation under the conditions of derivatization and/or high temperature liquid injection. This analysis revealed 224 features in all control and inoculated groups in total (Appendix A). Among these, 144 features could be assigned to trimethylsilyl (TMS) and methoxyamine (MeOX)-TMS derivatives of individual primary metabolites by spectral similarity search against an array of available libraries and/or co-elution with authentic standards. Some metabolites appeared as several MeOX-TMS isomers or derivatives with different numbers of TMS groups; therefore, the total number of identified (structurally annotated) metabolites appeared to be only 115 (summarized in Appendix A).

These metabolites represented 18 proteinogenic and eight non-proteinogenic amino acids, four amines, twelve fatty acids and esters, 17 organic acids, 22 sugars (monosaccharides including uronic, aldonic, aldaric acids, and phosphorylated derivatives, di- and oligosaccharides), ten polyols and their phosphorylated derivatives, ten phenolics, three heterocyclic compounds (nicotinic acid, 4-hydroxypyridine, uracil), three sterols and other triterpenes (cholesterol, β-sitosterol and trans-s18 proteinogenic qualene), and eight representatives of other classes.

The relative abundances of thermally labile metabolites were addressed by RP-IP-HPLC-QqQ-MS using a targeted metabolomics approach. This technique revealed in total 134 metabolites, which could be identified by co-elution with authentic standards (Appendix A). These metabolites were represented with 26 nucleosides and nucleotides and their derivatives; 23 amino acids; 29 sugars, sugar phosphates and their derivatives; 43 carboxylic acids; eight coenzyme-A derivatives; three inorganic compounds and two representatives of other classes. After merging the two result sets, a combined matrix with 358 entries was built and processed with the MetaboAnalyst 6.0 online software tool.

Comparison of all four groups by principal component analysis (PCA) did not show complete separation of all analyzed groups in the corresponding scores plot, while 47.9% of the total variance could be explained by the first principal component (Figure 5A). Hierarchical clustering analysis with a heatmap representation of the normalized relative abundances corresponding to individual metabolites revealed high dispersion in all experimental groups. This effect was most pronounced on the 22nd d.a.i. in the group grown in the presence of the agrobacterial suspension (Figure 5B).

Taking the results of the PCA analysis into account, we decided, when considering paired inter-group comparisons, to get better access in the tumor-associated dynamics of the plant metabolome. Following this logic, we first addressed age-related changes in the metabolome of radish calluses in the absence of inoculation with agrobacteria. For this, we compared the mock-treated plants (controls) harvested on the 10th and 22nd d.a.i. (Figure 6).

The PCA analysis revealed clear separation of the analyzed groups in the corresponding plots with 73.9% of the total variance explained by the first principal component (PC1) and 10.7% explained by principal component 2 (PC2) (Figure 6A). Despite the pronounced intra-group variance observed at the corresponding heatmap (Figure 6B), the *t*-test analysis with a volcano plot representation (FDR correction with the Benjamini–Hochberg method at *p* ≤ 0.05 and FC ≥ 2) yielded eight metabolites, which showed significantly decreased abundance and 16 metabolites, which were more abundant in the calluses cultured for 22 days in comparison to those grown for 10 days (Figure 6C, Table 2). Taking into account that some metabolites were detected by both LC-MS and GC-MS, the number of up-regulated metabolites dropped to 13.

The down-regulation pattern was dominated by amino acids, although their age-dependent decrease was moderate, i.e., 3.4-fold or lower (Table 2). The up-regulation patterns were mostly represented by intermediates of the central pathways (predominantly, tricarboxylic acid cycle, TCA) and metabolism of the nucleotides involved in energy production (Table 2).

Inoculation with the agrobacterial culture clearly affected the age-related metabolome dynamics of the *R. sativum* calluses. Thus, in contrast to the differences observed in the control group, the metabolomes of the calluses inoculated with *A. tumefaciens* showed no significant changes on the 22nd d.a.i. in comparison to the 10th d.a.i. (Appendix A). Indeed, the PCA analysis revealed the lack of separation of the analyzed groups in the corresponding scores plots with 50.1% of the total variance explained by PC1 and 20.2% explained by PC2 (Appendix A). Hierarchical clustering analysis with heatmap representation showed moderate intra-group variance (Appendix A). In agreement with this, the *t*-test analysis with a volcano plot representation (FDR correction with the Benjamini–Hochberg method at *p* ≤ 0.05 and FC ≥ 2) revealed no significant differences between 22 d.a.i. and 10 d.a.i. (Appendix A).

At the next step, we addressed the effect of agrobacterial infection on the metabolomes of the *R. sativum* calluses at each time point (i.e., 10 and 22 d.a.i.) separately. For this, we compared the patterns of primary metabolites detected in the radish calluses treated with agrobacterial culture with those detectable in the mock-treated samples.

On the 10th d.a.i., these patterns appeared to be clearly different. Indeed, the PCA analysis revealed good separation of the two analyzed groups in the corresponding scores plot with 63.7% of the total variance explained by the first principal component (Figure 7A). Hierarchical clustering analysis with a heat map representation of normalized relative abundances of individual metabolites showed that the extracts prepared from the inoculated and mock-treated calluses clustered separately (Figure 7B). The *t*-test analysis with a volcano plot representation (FDR correction with the Benjamini–Hochberg method at *p* ≤ 0.05 and FC ≥ 2) yielded 12 metabolites, which appeared to be more abundant in the agrobacteria-inoculated calluses after 10 days in comparison to the mock-treated controls (Figure 7C, Table 3).

The up-regulated metabolites included 2-phosphoglyceric acid, 3-phosphoglyceric acid (FC 2.2–2.3), myo-inositol phosphate (FC 2.3), as well as sugar phosphates (up to a 4.8-fold increase observed for fructose-1,6-diphosphate). The maximal increase of up to 5.1-fold was observed in the group of inoculated calluses on the 10th day for ADP-ribose.

Interestingly, the effect of the agrobacterial infection on the *R. sativus* calluses was much less pronounced at later steps of tumorigenesis (22nd d.a.i.). Indeed, as can be seen from the corresponding score plots, the PCA analysis did not reveal any separation between the inoculated and mock-treated groups, with only 43.7% of the total variance explained by the first principal component (Appendix A). In agreement with this, the hierarchical clustering analysis with heat map representation of the normalized relative abundances of individual metabolites revealed high intra-group variability in the inoculated callus groups on the 22nd d.a.i. (Appendix A). The comparison of the two groups by means of *t*-test analysis with a volcano plot representation (FDR correction with the Benjamini–Hochberg method at *p* ≤ 0.05 and FC ≥ 2) yielded no differentially abundant metabolites in the extracts prepared from the inoculated and mock-treated calluses (Appendix A). However, the relaxation of the parameters for the FDR correction (the use of *p* ≤ 0.1 instead of *p* ≤ 0.05 without any alterations in other parameters) revealed only one difference: namely, a 7.3-fold decrease in the contents of proline in the inoculated calluses in comparison to the corresponding controls.

## 4. Discussion

### 4.1. Agrobacterial Transformation and Tumour Formation Affect Metabolism of Radish Seedlings

The presented data clearly indicate that agrobacterial infection strongly affected the proteome of radish calluses, interfering with plant ontogenesis in a complex, but defined way. The observed changes were generally consistent with published data and can largely be explained by metabolic rearrangements associated with tumor formation. Most often, in the context of changes induced by agrobacterial inoculation, early (3–6 d.a.i.) and late events (more than 30 d.a.i) are distinguished [29,30]. The early response of the proteome is manifested by enhanced expression of the proteins involved in redox regulation and protein folding [31,32], whereas the late response involves a more or less profound restructuring of metabolism [33]. The time point corresponding to the 10th d.a.i. is so far poorly covered in the literature. However, this time point might represent the transition state of the inoculation-induced proteome dynamics, and requires, therefore, special attention in terms of general understanding of the mechanisms behind the metabolic adjustment triggered by agrobacterial infection. According to the described scenario, a pronounced increase in relative abundances of antioxidant proteins (peroxidases and oxidoreductases) was detected on the 10th d.a.i., although it was not as strong as that observed during the first two days after inoculation (according to the literature) [29,34].

Induction of redox enzymes generally supports the principal mechanisms already proposed for plant responses to agrobacterial infection and associated tumor formation. Thus, according to the current state of knowledge, the principal factors of the plant response to the agrobacterial infection are hypoxic and osmotic stress [35]. Hypoxic stress is associated with compromised aeration of the host tissues due to tumor growth. Tumor-induced osmotic deprivation is assumed, by its origin, to be close to drought stress and can be explained in terms of the so-called “gall-constriction hypothesis”. This hypothesis assumes that the phytohormone ethylene (which is overproduced due to the increase in the activity of 1-aminocyclopropane-1-carboxylate (ACC) synthase in response to higher tissue contents of auxins and cytokinins [36,37]) is involved in the establishment of highly vascularized tissues in the formed tumors at the detriment of the aerial part, leading to water-flow priority to tumor cells over the host shoot [36,38]. Another important aspect is that the growing tumor can damage epidermal structures, resulting in enhanced evaporation from the tumor surface.

Although generally our observations were in line with this scenario, some features of the agrobacteria-induced rearrangement in plant metabolism did not fit the described pattern of tumor-induced physiological responses. Thus, our data only partly support the involvement of the drought stress response in the observed metabolic shifts. Indeed, we could not detect any alterations in the relative abundances of the proteins associated with abscisic acid-mediated responses and no changes in the dynamics of drought-protective polypeptides (e.g., LEA proteins) or low-molecular weight osmolytes (glycine, betaine, choline, sucrose, raffinose family oligosaccharides) [35]. However, the involvement of tissue dehydration in triggering the observed metabolic rearrangements cannot be excluded. Thus, the tumor tissues featured higher relative abundances of fatty acid desaturases and several enzymes of glycan biosynthesis, which might be involved in maintaining plasma membrane integrity under dehydrated conditions [39,40].

### 4.2. The Tumour-Induced Metabolic Shifts Can Be Explained in the Context of Hypoxic Stress Response

The observed metabolic shifts can be much more easily explained in the context of the hypoxic stress response. Firstly, induction of peroxidases in inoculated calluses supports this assumption. The second important feature is tumor-induced suppression of photosynthesis and enhancement of glycolysis [41]. Finally, formation of tumors results in rearrangement of energy metabolism in the way that we see here—increase of ATP formation in glycolysis (Figure 8A) with simultaneous suppression of energy generation in the TCA cycle and oxidative phosphorylation (Figure 8B). These aspects are discussed in more detail below.

As far as the first aspect is concerned, significant suppression of the enzyme-based antioxidant system could be seen on the 22nd day after agrobacterial infection (Appendix A), following the initial oxidative burst induced by agrobacterial inoculation, which is well-described in the literature [29,42]. This observation is in good agreement with the scenario described in the literature. The latter assumes pronounced inhibition of the cellular ROS-detoxification system triggered by agrobacterial transformation. Most likely, this effect is associated with uncontrolled proliferation of cells in agrobacterial tumors, which might occur in the presence of large amounts of metabolically available nutrients [10]. This ultimately requires enhancement of all energy-related and biosynthetic metabolic processes, which is manifested by up-scaling of all cellular metabolic fluxes including ROS generation in electron-transporting chains (ETCs).

Such explosive enhancement of cellular metabolism and accompanied shifts in ROS homeostasis has already been described for mammalian tumors. Indeed, tumorigenesis is accompanied with enhanced ROS generation in mitochondrial ETCs, endoplasmic reticulum (ER) and NADPH oxidase system in comparison to non-cancer cells [43]. Moreover, overproduction of ROS can serve as a factor of tumorigenesis. Indeed, due to high potential of ROS for oxidative DNA damage [44,45], their overproduction might result in genomic instability, which essentially contributes to establishment of tumors [45].

Another important aspect of the tumor-induced metabolic adjustment is rearrangement of carbohydrate metabolism, which is clearly manifested in up-regulation of enzymes of sucrose biosynthesis both on the 10th and 22nd d.a.i. The key enzyme of this pathway—sucrose synthase—was shown to be involved in both sucrose synthesis and degradation. This protein is ubiquitous in plants and catalyzes the chemical reaction UDP-glucose + *D*-fructose←→UDP + sucrose [46,47]. Here, we found that both sucrose synthase and UDP-glucosyltransferase showed significant increase in relative contents (in comparison to the corresponding controls) on the 10th and 22nd day post-infection. These enzymes belong to the key elements of sugar signaling. For example, sucrose synthase promotes organogenesis of root nodules in legumes [48]. In this context, this activity might be involved in regulation of tumor formation upon the agrobacterium-mediated genetic transformation [32,49]. In agreement with this, another important actor in sugar signaling—UDP-glucose (which is a substrate of UDP-glycosyltransferase)—was reported to be involved in the response to pathogens [32,50,51]. Interestingly, we did not see any tumor-related alterations in the sugar signaling pathway at the level of metabolites, which, most likely, indicates high plasticity and lability of plant primary metabolism.

Finally, the pathways related to energy generation appeared to be part of the seedling metabolism, which showed the most pronounced proteome shifts related to agrobacterial inoculation and tumor formation. Among them, the tricarbon acid (TCA) cycle appeared to be the most strongly affected by agrobacterial transformation (although this effect could not be confirmed at the level of metabolome, Figure 8B). This was most obviously seen in the reaction of citrate formation, which is metabolized by cytosolic aconitase and cytosolic NADP+-dependent isocitrate dehydrogenase to generate α-ketoglutarate. Relative abundances of aconitate hydratase in inoculated calluses were more than 100-fold increased both on the 10th and 22nd d.a.i. (Appendix A). Aconitate hydratase (aconitase) catalyzes stereo-specific isomerization of citrate to isocitrate via the cis-aconitate intermediate, being a key enzyme of the TCA cycle. The up-regulation of isocitrate dehydrogenase was less pronounced (up to eight-fold on the 10th d.a.i.), although also clearly observable. The latter protein belongs to the key regulatory enzymes of the TCA cycle—it impacts the control of the TCA cycle by the mechanism of allosteric inhibition by ATP and NADH [52]. Thus, the up-regulation of this enzyme might indicate release of this inhibition due to some degree of energy depletion, i.e., decrease in relative amounts of reduced nicotinamide dinucleotides, which are essential cofactors of multiple biosynthetic reactions [53]. Importantly, the relative abundance of the following three enzymes in the cycle were decreased.

Remarkably, the observed changes in the callus proteome were not accompanied by any changes in the relative contents of individual metabolites. The observed pattern of TCA enzyme expression might be explained by tumor-specific adaptation manifested by enhanced synthesis of amino acids, primarily arginine as the precursor of opines.

The absence of changes in the dynamics of corresponding metabolites might indicate that their equilibrium tissue concentrations were stable not only in the mock-treated seedlings, but also upon tumor induction. The fact that no TCA intermediates accumulate under these conditions might indicate high efficiency of this tumor-specific adaptation.

Generally, acetyl-co-enzyme A (acetyl-CoA)—a metabolic precursor of citrate, representing an important metabolic crossroad in the cell—connects the TCA cycle and oxidative phosphorylation in mitochondria with glycolysis and fatty acid synthesis in the cytosol [54]. In this study, we observed pronounced enhancement of glycolysis with multiple enzymes of this pathway slightly (approximately 1.5–3-fold), but significantly, up-regulated on the 22nd day of the experiment (Figure 8A, Appendix A). The only exclusion was glyceraldehyde-3-phosphate dehydrogenase, which was enhanced by two to three orders of magnitude at both sampling points (Appendix A). This protein is a key enzyme in glycolysis that catalyzes the first step of the pathway by converting *D*-glyceraldehyde 3-phosphate (G3P) into 3-phospho-*D*-glyceroyl phosphate (glycerate-1,3-diphosphate) [55]. This enzyme is also involved in intra-nuclear interactions, and impacts on nuclear events including transcription, RNA transport, DNA replication and apoptosis [56].

Remarkably, all these enzymes (2,3-bisphosphoglycerate-independent phosphoglycerate mutase 1, pyruvate kinase, enolase, fructose-bisphosphate aldolase and glyceraldehyde-3-phosphate dehydrogenase) represented the second half of the glycolytic pathway, whereas the first part was down-regulated (Appendix A). This protein dynamics pattern matched well to the metabolomics data (Table 3), which indicated accumulation of glycolytic metabolites (especially of the first part of the pathway—glucose-6-phosphate, fructose-6-phosphate, fructose-1,6-diphosphate) and metabolites of nucleotides involved in energy metabolism by the 10th d.a.i. The latter represent macroergic phosphorus-containing compounds—sugar phosphates, nucleophosphates and phosphorus-containing acids (Table 3). These compounds are necessary for multiple biochemical processes and play an exceptional role as energy carriers in cell protein syntheses important for growth, in carbohydrate metabolism, in photosynthesis and respiration [57]. Thus, this co-accumulation of the macroergic nucleotide precursors and sugar phosphates might indicate activation of a less efficient energy production route—anaerobic glycolysis (Figure 8B).

It can be assumed that even a minimal increase in relative abundances of glycolytic enzymes might essentially enhance generation of acetyl-CoA by the pyruvate dehydrogenase complex and its involvement in the TCA cycle (Figure 8). Thus, here we could observe a concerted increase in the expression of glycolytic, TCA- and TCA-associated enzymes, which are located in this pathway up-stream relative to α-ketoglutarate. Remarkably, expression of the enzymes downstream of the latter metabolite (succinate dehydrogenase and transketolase) was slightly (approximately 1.5-fold) suppressed. It can be assumed that the overall above-described expression shift represented the biochemical manifestation of the metabolic adaptation directly related to the agrobacterial infection. Indeed, most likely, the observed proteome dynamics can be explained by the continuous removal of alpha-ketoglutarate, a substrate of *L*,*L*-diaminopimelate aminotransferase, which yields glutamate, a precursor of arginine and octopine [58].

This assumption prompted us to analyze the shifts in amino acid metabolism in more detail. As can be seen from Table 3, no changes in relative abundances of individual intermediates of amino acid biosynthetic pathways could be observed at either sampling point. This can be explained by fast metabolite conversion at each enzymatic step and rapid metabolic fluxes throughout corresponding pathways. However, long-term adaptation at the level of the proteome could be clearly seen. Thus, a slight increase in arginine synthesis could be observed by the 10th day, while a significant decrease in the relative abundance of glutamine synthase (i.e., enzyme which catalyzes biosynthesis of glutamine—the precursor of arginine) could be observed on the 22nd day. This expression dynamic might indicate a predominantly anabolic role of the TCA cycle under the conditions of agrobacterial transformation, which are accompanied with synthesis of specific amino acids and opiates.

### 4.3. Age-Related Dynamics of Seedling Proteome and Metabolome Are Strongly Affected by the Agrobacterial Infection and Tumour Formation

Comparison of the tumor-induced metabolic shifts (Appendix A and Table 3) with those induced by constitutive ontogenetic changes (Appendix A and Table 2) allowed the making of several important observations. Thus, expression of the central metabolism pathways increased over the twelve days between the two sampling points. However, in contrast to the tumor-induced changes, age-related changes in the calluses’ primary metabolism are balanced, i.e., both the first and second parts of glycolysis and the TCA cycle are up-regulated. This indicates continuous increase of energy production over the period of vegetative growth, whereas cellular energetic activity is suppressed under tumor conditions. Further, under the conditions of non-tumorous development, TCA acidic intermediates accumulate due to continuous increase of the metabolic flux via the TCA cycle (Table 2). This underlies constitutive age-dependent enhancement of amino acid metabolism and rich patterns of age-related dynamics of bin 29 (protein metabolism). Thereby, both branches—protein biosynthesis and degradation—are balanced (Appendix A).

Accumulation of amino acids under non-stressed conditions might be of high physiological relevance. Increasing the level of proline (four-fold) enhanced the immunity of plants under stress conditions, the accumulation of nitrogen, and regulates water exchange in the plant [59,60,61]. Aconitic acid (a 3.5-fold increase on the 22nd d.a.i. in comparison to the 10th day) maintains the oxidation–reduction balance, and protects against pests and diseases [62]. An increase in the content of organic acids of the tricarboxylic acid cycle—fumaric acid (2.5 times), citric acid (4.6 times) and iso-citric acid (4.9 times)—may be associated with the participation of organic acids in osmoregulation [63]. Under the same conditions, metabolites with a reduced content (2–3.4-fold) on day 22 (down-regulated) includes amino acids—Phe, Ser, Gly, Leu, Ile, Asn, Val—that play a fundamental role in plant metabolism, development and the transmission of cellular signals [64]. Under the influence of stress factors, plants begin to use their accumulated internal resources against the background of a slow metabolism. The longer the exposure to adverse conditions, the more amino acid deficiency is felt [65].

It is important to mention that reduced flux over the second half of the TCA cycle ultimately reduces its role as the energy source. In good agreement with this consideration, several important energy-consuming cellular proteins were affected by agrobacterial infection and tumor establishment.

Thus, transport function is obviously affected by agrobacterial infection: both on the 10th and 22nd days after inoculation there is a decrease in the expression of vacuolar-type ATPase (V-ATPase) (Appendix A). This is a highly conserved evolutionarily ancient enzyme that uses the energy of ATP hydrolysis to create a proton gradient. V-ATPases link the energy of ATP hydrolysis to proton transport across intracellular and plasma membranes of eukaryotic cells [66]. One can assume that such changes might compromise such fundamental processes as cellular nutrition, excretion, inter-cellular signaling, cell wall synthesis and stability.

Further, a decrease in the expression of cyclin D3 involved in the induction of mitotic cell division [67] was observed at both monitored stages of tumor formation (Appendix A). This might indicate a deleterious effect of agrobacterial infection on cytoskeleton functions and cell division. Along with fundamental rearrangement of amino acid biosynthesis in favor of specific amine metabolites, pronounced decrease in relative abundances of the nucleotide biosynthesis pathways, transcriptional and protein biosynthesis machinery could be observed. This is well-illustrated by the fact that the numbers of the proteins, down-regulated by the 10th d.a.i., are three times higher in comparison to the numbers of those up-regulated at this time point (Appendix A). The second sampling point (22nd d.a.i.) is characterized by the highest numbers and a greater diversity of proteins, both in up- and down-regulated groups. On day 22, the trend for domination of down-regulation in the protein expression patterns continues. Most of the differentially abundant proteins belonged to the groups related to stress, redox processes and protein and amino acid metabolism.

Most likely, this is a result of re-distribution of available cellular resources in favor of the accumulation of specific amino acids and their characteristic derivatives (this appears to be likely with respect to octopine synthesis). It is important to note that accompanying suppression of translation is accompanied with a parallel decrease in the levels of transcription. It is possible to assume that these rearrangements are mediated by sugar signaling. Indeed, on the 22nd d.a.i., relative abundances of cell wall invertases—key players of this pathway, as well as sucrose synthase and fructokinase—regulators of autophagy, increase. Another important observation, made on the 22nd d.a.i., is a clear increase in the expression rates of several glycolytic enzymes (e.g., phosphofructokinase and pyruvate kinase). There is also an increase in the relative abundance of malate dehydrogenase, which finally leads to the conclusion that the tricarboxylic acid cycle has a predominantly catabolic function under these conditions. Switching off the anabolic function of the CTC is accompanied by a decrease in proteins abundant in the previous step—phosphoenolpyruvate carboxylase (PEPC), as well as enzymes of amino acid and lipid biosynthesis.

Simultaneously with those changes, protein synthesis is activated—the content of a number of ribosomal proteins is increased (renewal of the translational apparatus takes place). An increase in the enzyme content of cell wall polymer hydrolysis indicates growth by stretching—another energy-consuming process. Destruction of other ribosomal proteins is accompanied by an increase in relative abundances of individual proteasome subunits. In agreement with the data available in the literature [68], pronounced activation of the antiglycative defense was observed in this study—the content of the enzyme phosphoribulokinase was increased. Also, a decrease in the content of proteins of antioxidant protection—peroxidase, glutathione S-transferase and others—was observed. The relative nature of the changes in the expression of photosynthesis proteins remains somewhat unclear, although a general downward trend in its intensity can be seen. Thus, the second phase of tumor development is characterized by more balanced energy metabolism and biosynthesis, lower stress levels, higher activity of protein-degrading systems and more active growth.

### 4.4. Effect of Agrobacterial Transformation on the Tumor- and Meristem-Related Peptides and Proteins of Radish Seedlings

In addition to full-proteome profiling and analysis of differential protein expression patterns, we performed targeted analysis of several proteins of the conserved WOX-CLAVATA regulatory system based on a defined search for targeted MS/MS predicted proteolytic peptides. In plants, CLAVATA3/EMBRYO SURROUNDING REGION-RELATED (CLE) genes encode a large family of extracellular signaling peptides that stimulate receptor-mediated signal transduction cascades to modulate various morphogenesis processes [69]. We showed that the expression of CLE peptides was consistent between experimental points. Thus, the protein with increased expression level in the presence of *Agrobacterium* infection only at day 10 is CLAVATA3/ESR-RELATED 22, while CLAVATA3/ESR-RELATED 45 shows increased expression level at both day 10 and day 22. The function of CLE22 is known to be regulation of tap root growth through an auxin signaling-related pathway in radish (*Raphanus sativus* L.) [70]. In addition, in *Arabidopsis thaliana*, CLE22 is involved in intercellular signaling related to cell fate determination and in maintaining root meristem identity [71]. CLE45 also regulates cell fate by inhibiting the maintenance of the root apical meristem and preventing the response to auxin in the root meristem, resulting in suppression of protofloem differentiation and cell division of pericladial sieve precursor cells [72].

CLAVATA3/ESR-RELATED 5, the transcription level of which is differentially regulated by the phytohormone auxin, showed increased expression levels at day 22 after inoculation [73].

A wide range of functions of the GRAS family transcription factor, which in addition to participating in the formation of the apical and axillary meristem, radial root patterning and the signaling pathway of phytohormones (gibberellins), plays a key role in the organization of symbioses and stress response, is the reason for the appearance of a peptide of this family in the cluster associated with a decrease in expression level for 10th d.a.i. and another increasing expression for 22nd d.a.i. [74].

The function of cyclin D3, which also demonstrates a decrease in expression level on day 10, is to regulate cell division, specifically, the control of the transition from the G1 to S phase [75,76]. Probably, in this case, its decrease indicates a decrease in mitotic activity, since it is known that the activity of cyclin D3 is maximal during the transition from the G1 to the S stage of the cell cycle, and its expression subsequently decreases [76]. A decrease in cyclin D3 levels may also be associated with a high level of oxidative stress, the development of which is evidenced by the proteomic data we have obtained.

## 5. Conclusions

Infection of higher plants with the bacterium *Agrobacterium tumefaciens* (a causative agent of crown gall disease) represents one of the most comprehensively characterized examples of plant–microbial interactions. However, although genetic aspects of this interaction have been comprehensively described to date, the accompanying shifts in plant metabolism are still poorly characterized. Moreover, only limited information on the proteome responses induced by agrobacterial infection is available. Thus, our study is the first LC-based comprehensive proteomics survey of a plant’s response to formation of agrobacteria-induced tumors. This approach allowed reliable characterization of the molecular mechanisms behind these responses. We succeeded in revealing rearrangement of energy metabolism and its switch from oxidative to anaerobic metabolism as a part of plant adaptation to tumor-induced hypoxic stress. Thus, it was found that by the 10th day after inoculation, a decrease in the role of aerobic pathways and an increase in the role of anaerobic pathways of energy production and activation of sugar signaling were observed. When considering the tumor-induced changes in the ontogenetic context, an increase in protein metabolism (with pronounced predominance of protein biosynthesis over its degradation) could be observed by the 22nd day after inoculation. However, to get better insight into the dynamics of this response, further time points before the 10th d.a.i. need to be addressed.

## Figures and Tables

**Figure 1 biomolecules-15-00290-f001:**
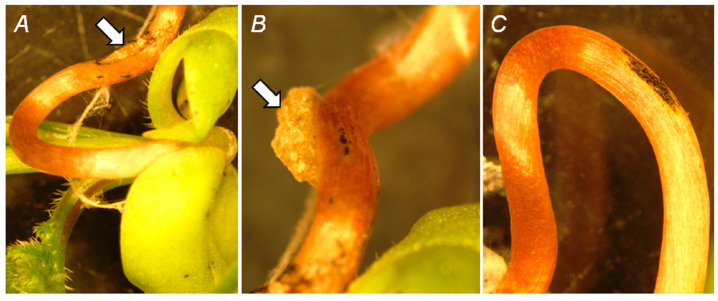
Hypocotyls of radish (*Raphanus sativus*) seedlings on the 10th (**A**) and 22nd (**B**) days after inoculation (d.a.i.) with *Agrobacterium tumefaciens.* White arrows indicate crown galls. (**C**) represents the result of the mock treatment (cultural medium) observed on the 10th d.a.i.

**Figure 2 biomolecules-15-00290-f002:**
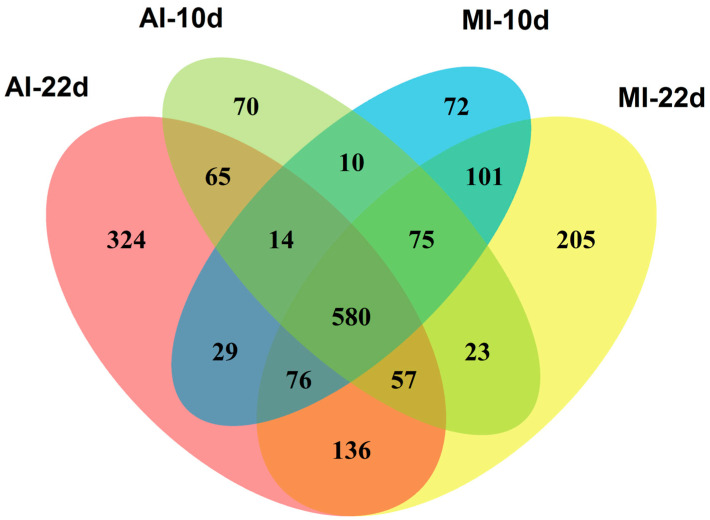
The numbers of non-redundant proteins (protein groups) identified in radish (*Raphanus sativus*) plants on the 10th (10d) and 22nd (22d) days after inoculation (d.a.i.) with cultural medium (MI) and *Agrobacterium tumefaciens* culture (AI). The tryptic digests (*n* = 3), obtained from radish seedlings, were analyzed by nano-high performance liquid chromatography-electrospray ionization linear ion trap-orbital trap mass spectrometry (nanoHPLC-ESI-LIT-Orbitrap-MS) in DDA mode.

**Figure 3 biomolecules-15-00290-f003:**
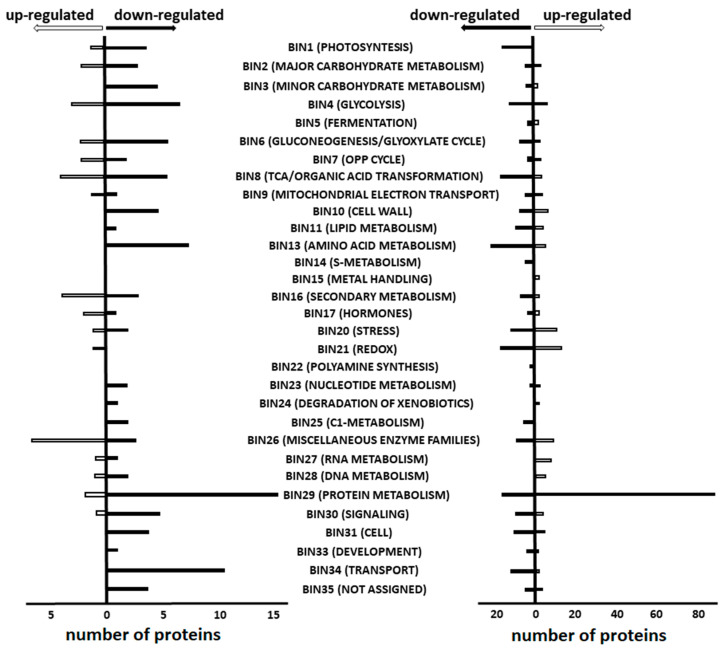
Functional annotation of the proteins, differentially expressed in *R. sativus* seedlings on the 10th (left) and 22nd (right) day after inoculation (d.a.i.) with cultural medium and suspension of *A. tumefaciens*. White and black boxes denote the proteins, up- and down-regulated, respectively, in the plants inoculated with *A. tumefaciens* culture in comparison to those inoculated with cultural medium. Functional annotation relied on MapMan annotation with subsequent manual curation of data based on the literature and database entries.

**Figure 4 biomolecules-15-00290-f004:**
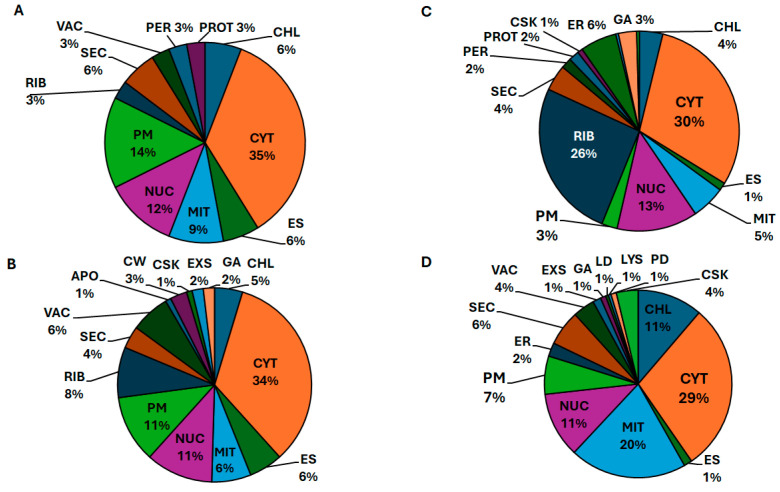
Prediction of intracellular localization for the differentially expressed proteins, annotated in isolates, obtained from *Raphanus sativus* L. plants, up-regulated (**A**,**C**) and down-regulated (**B**,**D**) on the 10th (**A**,**B**) and 22nd (**C**,**D**) day after inoculation (d.a.i.) with suspension of *Agrobacterium tumefaciens* in comparison to cultural medium. The prediction relied on the BUSCA tool and protein sequences from combined *Raphanus sativus* database (NCBI, Uniprot and 2015 *R. sativus* genome assembly), and was verified manually afterwards—the procedure relied on the accession-based search against free databases: UniprotKB (https://www.uniprot.org/), nextprot database (https://www.nextprot.org/), BRENDA Enzyme Database—BRENDA (https://brenda-enzymes.org/). APO—apoplast; CHL—chloroplast; CSK—cytoskeleton; CW—cell wall; CYT-cytoplasm; ECS—extracellular space; ER—endoplasmic reticulum; ES—endomembrane system; GA—Golgi apparatus; LD—lipid droplet; LYS—lysosome; MIT—mitochondrion; NUC—nucleus; PD—plasmodesma; PER—peroxisome; PL—plasma membrane; PROT—proteasome; RIB—ribosome; SEC—secreted; VAC—vacuole.

**Figure 5 biomolecules-15-00290-f005:**
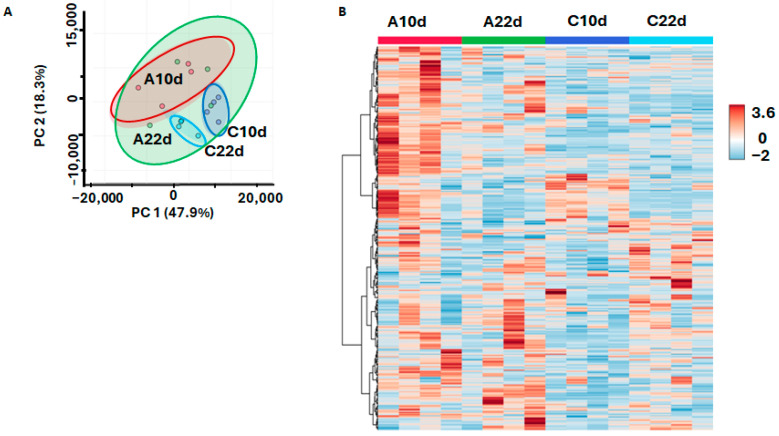
Comparison of the primary metabolite profiles detectable in aq. methanolic extracts prepared from the *R. sativus* calluses. The calluses were harvested on the 10th and 22nd days after inoculation (d.a.i.) with the *Agrobacterium tumefaciens* suspension (A10d and A22d, respectively) or cultural medium (control, mock treatments, C10d and C22d, respectively): results of the principal component analysis (PCA) with a score plot representation (**A**), hierarchical clustering analysis with a heatmap representation (**B**).

**Figure 6 biomolecules-15-00290-f006:**
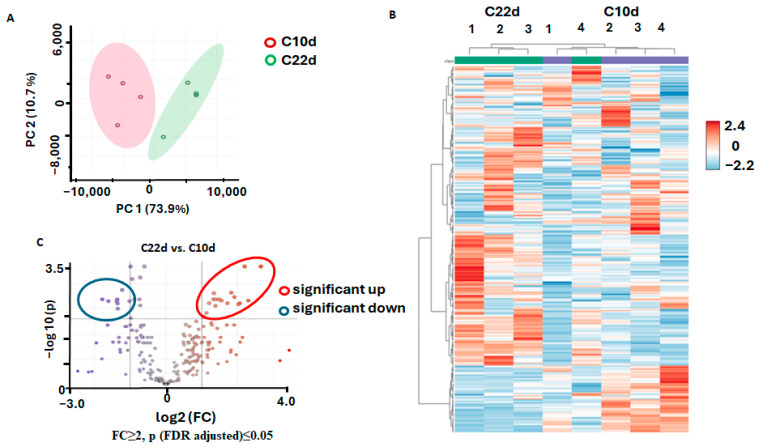
Comparison of the primary metabolite profiles detectable in aq. methanolic extracts prepared from the *R. sativus* calluses, cultivated on the bacteria-free cultural medium on the 10th day—C10d and on the 22nd day—C22d. Statistical analysis relied on principal component analysis (PCA) with a score plot representation (**A**), hierarchical clustering analysis with a heatmap representation (**B**) and volcano plot with a graphical representation of differentially abundant analytes—(**C**) with Benjamini–Hochberg false discovery rate (FDR) correction at *p* ≤ 0.05 and fold change (FC) ≥ 2. Colored dots indicate metabolites showing statistically significant differences with FC ≥ 2 threshold level at *p* ≤ 0.05 compared to control; that is, red dots indicate metabolites with the contents higher on 22nd day in the control in comparison to the control 10th day ones. The metabolites marked with grey dots showed no statistically significant differences.

**Figure 7 biomolecules-15-00290-f007:**
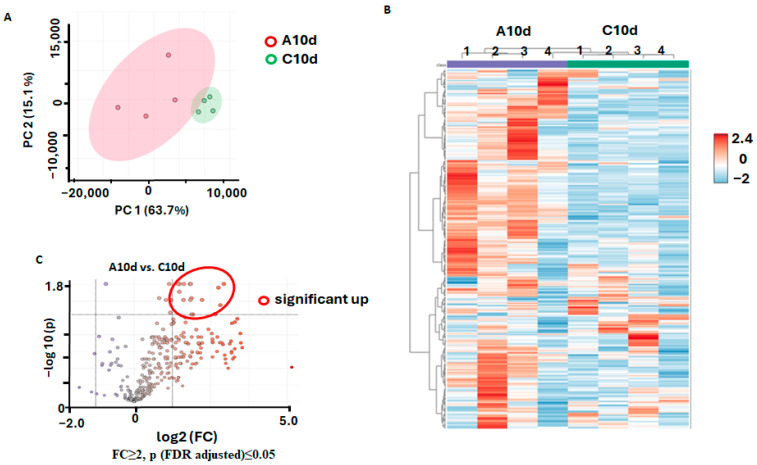
Comparison of the primary metabolite profiles detectable in aq. methanolic extracts prepared from the *R. sativus* calluses. The calluses were harvested on the 10th day after inoculation with a suspension of the *A. tumefaciens* culture (A10d) or corresponding mock-treated controls (C10d—exposure to the bacteria-free cultural medium). Statistical analysis relied on the principal component analysis (PCA) with a score plot representation (**A**), hierarchical clustering analysis with a heatmap representation (**B**) and volcano plot with a graphical representation of differentially abundant analytes—(**C**) with Benjamini–Hochberg false discovery rate (FDR) correction at *p* ≤ 0.05 and fold change (FC) ≥ 2. Colored dots indicate metabolites showing statistically significant differences with FC ≥ 2 threshold level at *p* ≤ 0.05 compared to control; that is, red dots indicate metabolites with the contents higher in inoculated calluses on the 10 d.a.i. in comparison to the control ones. The metabolites marked with grey dots showed no statistically significant differences.

**Figure 8 biomolecules-15-00290-f008:**
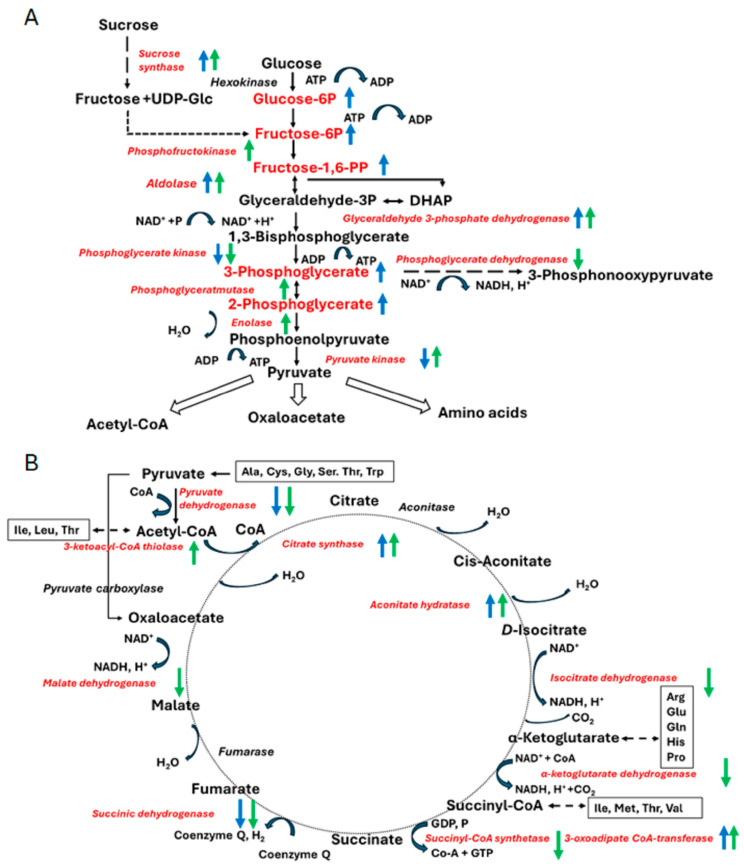
Changes in the glycolysis (**A**) and tricarbon acid cycle (**B**) pathways observed in the calluses on the 10th and 22nd day after inoculation (d.a.i.) of radish seedlings with the *Agrobacterium tumefaciens* culture. Blue and green arrows denote statistically significant expressional/abundance differences observed on the 10th and 22nd d.a.i. (in comparison to the corresponding mock controls), respectively, with upwards and downwards oriented arrows for up- and down-regulated proteins/metabolites, respectively. Individual proteins and metabolites demonstrating significant changes in comparison to the corresponding mock controls are marked in red.

**Table 1 biomolecules-15-00290-t001:** Tumor- and meristem-related peptides and proteins differentially expressed in the leaf of *R. sativus* seedlings after treatment with the suspension of *A. tumefaciens* and bacteria-free culture medium (mock treatment).

ProteinClusters(10d/22d) *^a^*	№	Proteins	log2FCs	*p* *^d^*	*q*
Accession	Description *^b^*	Function *^c^*	R-10d	R-22d	R-MI	R-AI
Cluster 1(up/steady) 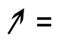	1	NP_680162.1	CLAVATA3/ESR-RELATED 22	development	1.9	NS	NS	NS	2.00 × 10^−4^	9.00 × 10^−3^
Cluster 2(up/up) 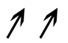	2	NP_001077799.1	CLAVATA3/ESR-RELATED 45 ***^e^***	developmentauxin signaling	1.5	1.06	1.2	1.8	1.16 × 10^−3^/1.86 × 10^−6^*(6.82 × 10^−3^*/*4.61 × 10^−4^)*	2.12 × 10^−2^/1.00 × 10^−4^*(2.73 × 10^−2^*/*3.90 × 10^−4^)*
Cluster 3(down/steady) 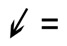	3	NP_566783.1	CLAVATA3/ESR-RELATED 27	development	0.74	NS	0.87	1.92	1.81 × 10^−3^*(2.45 × 10^−3^*/*1.62 × 10^−5^)*	2.12 × 10^−2^*(1.47 × 10^−2^*/*3.11 × 10^−5^)*
4	NP_190576.1	CYCLIN D3;3	cell division	0.9	NS	0.74	0.7	1.00 × 10^−2^*(9.00 × 10^−3^*/*5.00 × 10^−2^)*	4.00 × 10^−2^*(3.00 × 10^−2^*/*3.00 × 10^−3^)*
5	NP_191622.1	GRAS family transcription factor	transcription	0.66	NS	NS	0.81	3.38 × 10^−3^*(1.47 × 10^−2^)*	2.72 × 10^−2^*(1.47 × 10^−3^)*
Cluster 4 (steady/up) 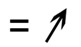	6	NP_195389.4	GRAS family transcription factor	transcription	NS	1.74	NS	0.8	1.39 × 10^−5^*(1.49 × 10^−2^)*	3.76 × 10^−4^*1.47 × 10^−3^)*
7	NP_850159.2	CLAVATA3/ESR-RELATED 5	development	NS	2.4	NS	NS	2.17 × 10^−4^	4.00 × 10^−3^
Cluster 5(steady/steady) 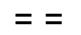	8	NP_001318877.1	CLAVATA3/ESR-RELATED 21 ***^e^***	development	NS	NS	1.9	NS	*1.66 × 10^−7^*	*1.99 × 10^−6^*
9	NP_173493.2	WUSCHEL-related homeobox 14 ***^e^***	developmentDNA-binding	NS	NS	NS	1.09	*5.00 × 10^−2^*	*9.00 × 10^−2^*
10	NP_187735.2	WUSCHEL related homeobox 5 ***^e^***	developmentDNA-binding	NS	NS	0.4	1.65	*9.53 × 10^−3^*/*1.00 × 10^−3^*	*2.86 × 10^−2^*/*3.00 × 10^−4^*
11	NP_197904.1	homeobox protein knotted-1-like 3 ***^e^***	DNA-binding	NS	NS	NS	1.32	*2.2 × 10^−2^*	*1.9 × 10^−3^*
12	NP_683589.1	CLAVATA3/ESR-RELATED 19 ***^e^***	development	NS	NS	0.93	NS	*2.00 × 10^−2^*	*3.00 × 10^−2^*
13	NP_001319370.1	CLAVATA3/ESR-RELATED 1 ***^e^***	development	NS	NS	1.22	NS	*2.00 × 10^−2^*	*3.00 × 10^−2^*
14	NP_195142.1	CYCLIN D3;1	cell divisionmitosis	NS	NS	NS	1.45	*2.71 × 10^−2^*	*2* *.08 × 10^−3^*
15	NP_195014.1	homeobox-leucine zipper protein ATHB-8 ***^e^***	development	NS	NS	NS	0.71	*2.3 × 10^−2^*	*1.99 × 10^−3^*

***^a^*** Clusters were defined in the paired comparisons of relative protein abundances in the groups of radish plants inoculated with *A. tumefaciens* (AI) and bacteria-free cultural medium (MI). The proteins were isolated on the 10th and 22nd d.a.i., hydrolyzed with trypsin and resulted digests were analyzed by nanoHPLC-ESI-LIT-Orbitrap-MS. The direction of the protein expression changes, i.e., up (↑)- and down (↓)-regulation or absence of expressional changes (steady =), was derived from relative abundance ratios (AI/MI), assessed on the 10th (10d) and 22nd (22d) days after inoculation. ***^b^*** Descriptions of individual protein were extracted from the headers of corresponding fasta files. ***^c^*** Protein functions are given according to UniprotKB. Binary logarithm of fold changes (log_2_FCs) were calculated for the protein abundance ratios AI-10d/MI-10d (10d), AI-22d/MI-22d (22d), MI-22d/MI-10d (MI) and AI-22d/AI-10d (AI); ***^d^***
*p* values were obtained by one-way ANOVA using Progenesis QI software; the *q* values were obtained with Progenesis QI software; ***^e^*** proteins identified in a targeted search with a sequence database containing 37 selected peptides and proteins (Appendix A) and manually checked for the quality of identification (Appendix A); NS—“Non-significant” denotes non-confident hits or the changes below 1.5-fold in absolute scale or <0.6 and >−0.6 in log_2_ scale; additional *p* and *q* values in italics denote that these proteins were also found in MI-22d/MI-10d (MI) and AI-22d/AI-10d (AI) groups.

**Table 2 biomolecules-15-00290-t002:** Primary metabolites of *R. sativus* calluses demonstrating statistically significant up- and down-regulation in control on the 22nd day in comparison to the 10th day.

Metabolite	Regulation	Method	FC	log_2_ (FC)	*p*.adjusted *	−lg (*p*)
phenylalanine	down	GC-MS	2.0	1.0173	0.0280	1.5525
serine	down	GC-MS	2.3	1.2216	0.0144	1.8430
glycine	down	GC-MS	2.5	1.3449	0.0410	1.3869
leucine	down	GC-MS	2.5	1.3500	0.0319	1.4960
isoleucine	down	GC-MS	2.6	1.3602	0.0251	1.6007
asparagine	down	GC-MS	2.6	1.3742	0.0281	1.5508
valine	down	GC-MS	3.1	1.6110	0.0280	1.5525
isoleucine	down	GC-MS	3.4	1.7768	0.0251	1.6007
RI2117_Unknown	up	GC-MS	2.2	1.1302	0.0471	1.3268
malic acid	up	GC-MS	2.3	1.2051	0.0313	1.5039
fumaric acid	up	LC-MS	2.5	1.3250	0.0251	1.6007
adenosine triphosphate	up	LC-MS	2.5	1.3402	0.0167	1.7766
gluconic acid	up	GC-MS	2.6	1.3798	0.0360	1.4437
malic acid	up	LC-MS	2.6	1.3906	0.0251	1.6007
fructose-1,6-diphosphate	up	LC-MS	2.8	1.4813	0.0251	1.6007
galactonic acid/gluconic acid	up	LC-MS	2.9	1.5483	0.0278	1.5560
hexacosanoic acid	up	GC-MS	3.3	1.7153	0.0313	1.5039
aconitic acid	up	LC-MS	3.5	1.8223	0.0158	1.8009
citric acid	up	GC-MS	3.8	1.9073	0.0204	1.6907
proline	up	GC-MS	4.0	2.0165	0.0333	1.4777
adenosine diphosphoribose	up	LC-MS	4.2	2.0625	0.0313	1.5039
citric acid	up	LC-MS	4.6	2.1948	0.0053	2.2786
isocitric acid	up	LC-MS	4.9	2.2852	0.0280	1.5525
2,3-butanediol	up	GC-MS	6.3	2.6465	0.0053	2.2786

The confidence of the differences was * confirmed by *t*-test (*p* ≤ 0.05 with FDR correction) and fold-change (FC ≥ 2).

**Table 3 biomolecules-15-00290-t003:** Primary metabolites of *R. sativus* calluses demonstrating statistically significant agrobacterial-dependent up-regulation on the 10th d.a.i. in comparison to the corresponding age-matched controls.

Regulated Metabolites	Method	FC	log_2_ (FC)	*p* _adjusted_	−lg (*p*)
2-phosphoglyceric acid	LC-MS	2.2	1.1471	0.0175	1.7577
myo-inositol phosphate	GC-MS	2.3	1.1809	0.0175	1.7577
3-phosphoglyceric acid	LC-MS	2.3	1.1835	0.0254	1.5959
gluconic acid	GC-MS	2.5	1.2999	0.0175	1.7577
adenosine monophosphate	LC-MS	2.8	1.4674	0.0175	1.7577
glucose-6-phosphate	GC-MS	2.8	1.4710	0.0321	1.4934
2′-deoxyguanosine 5′-monophosphate	LC-MS	2.8	1.4779	0.0175	1.7577
fructose-6-phosphate/glucose-6-phosphate	LC-MS	2.8	1.4977	0.0310	1.5083
xanthosine-5′-phosphate	LC-MS	3.4	1.7757	0.0321	1.4934
fructose-1,6-diphosphate	LC-MS	4.8	2.2533	0.0383	1.4171
adenosine5′-diphosphoribose	LC-MS	5.1	2.3449	0.0175	1.7577

The confidence of the differences was confirmed by *t*-test (*p* ≤ 0.05 with FDR correction) and fold-change (FC ≥ 2).

## Data Availability

The raw data are available in the PRIDE repository under the project accession number PXD060463 and project DOI 10.6019/PXD060463. The other information (i.e., any processed data/software outputs) are at the authors’ disposal and are available by request.

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
