# Peer review of "Proteome and Metabolome Alterations in Radish (Raphanus sativus L.) Seedlings Induced by Inoculation with Agrobacterium tumefaciens"

_biomolecules, 2025, doi:10.3390/biom15020290_

Round 1
Reviewer 1 Report
Comments and Suggestions for Authors
The present article entitled “Alterations in the proteome of radish (Raphanus sativus L.) Seedlings induced by inoculation with Agrobacterium tumefaciensd” basd on an interesting theme and cover a broad aspect of interactions between Agrobacterium tumefaciens and higher plants , how the T-DNA of A. tumefaciens intgradation results in the tumor formation that results in changes in proteome and metabolome. The results indicate that A. tumefaciens infection induces complex metabolic reprogramming in plants, which may be aimed at optimizing energy use and supporting the changed growth patterns of the model plants associated with tumor formation.
The experiment is well designed and the presenatation of the article is looking fine.
I have a minor suggestion to just improve the quality of the Fig 6.c and Fig 7.c
Author Response
We thank the Reviewer for their thoughtful review and highly appreciate the valuable
comments and suggestions to improve the manuscript. Following these advices we performed
all required changes in corresponding sections, as indicated in the following rebuttal
addressing each aspect. These changes are referred with the line numbers which are also
highlighted as comments in the text
Reviewer 1
Remark
I have a minor suggestion to just improve the quality of the Fig 6.c and Fig 7.c
Answer
The issues are corrected (see the corresponding figures in the corrected version)

Reviewer 2 Report
Comments and Suggestions for Authors
This manuscript reports on the response of radish proteomic dynamics to Agrobacterium infection, with a focus on studying the proteomic changes associated with tumor development and the accompanying changes in primary metabolites. Although this study is interesting, there are still the following shortcomings and issues:
1. The title lacks a description related to metabolomics.
2. The number marked after the author's name should be before the comma.
3. The abstract lacks a description of the main research results.
4. The references cited in the preface do not include those from 2022-2024, do not mention metabolomics related studies, and do not mention the correlation analysis between proteomics and metabolomics. The result analysis also does not include the correlation analysis between proteomics and metabolomics.
5. The statement 'On the 10th and 21st days after inference' mentioned in line 141 contradicts the 10th and 22nd days in the results analysis.
6. The RNA isolation mentioned in line 143 did not yield any relevant results in the result analysis.
7. The header of Table 3 is quite strange, pay attention to where some bold words are located.
8. “rpm” should be converted to “g”.
9. Part of the fonts in Figure 3 are blurry or even ghosted, and Figure 5 is not clear enough.
10. The line numbers of lines 677 and 678 are not on the right side, blocking some of the content.
11. Some abbreviations that appear for the first time do not have their corresponding full names, such as OD and CV (is CV ≤ 60 correct?) FC, RI, and RSD. In addition, 'GDM' should be 'GMD'.
12. In lines 135 and 162, there is no space between the numbers "28oC" and "4oC" and their units, while in line 159, there is a space between the numbers "37 oC" and their units. Similarly, please check for similar issues.
13. Whether the numbering, first line suspension, and blank space before and after paragraphs of the secondary headings under different primary headings are inconsistent, please check the formatting of the entire text.
14. Please check if "2792" in line 299 is correct.
15. What is the basis for dividing proteins into 5 clusters in Table 1? Can the width of Table 1 be further widened to make it more aesthetically pleasing?
16. In the discussion, relevant model of protein and metabolite changes should be summarized in figure to reflect the rearrangement of energy metabolism that is favor of anaerobic pathways expressed in the abstract.
17. The subheading on line 746 is neither numbered nor formatted differently from the subheading on line 971, and is 4 pages long. It is recommended to divide it into several subheadings.
18. On line 1001, the reference number is missing in square brackets.
19. For other questions, please refer to the annotations in the PDF.

Finally, please revise the language expression of the manuscript.
Author Response
We thank the Reviewer for their thoughtful review and highly appreciate the valuable comments and suggestions to improve the manuscript. Following these advices we performed all required changes in corresponding sections, as indicated in the following rebuttal addressing each aspect. These changes are referred with the line numbers which are also highlighted as comments in the text
Reviewer 2
Remarks
Remark 1
The title lacks a description related to metabolomics.
Answer
We agree with the reviewer and extend the title as suggested. Indeed, despite the fact that our manuscript considers mostly changes in seedling proteome, accompanying alterations in seedlings metabolome were also addressed there (although in lower extent). Please, see new title – “Proteome and metabolome alterations in radish (Raphanus sativus L.) seedlings induced by inoculation with Agrobacterium tumefaciens” (lines 1-4)
Remark 2
The number marked after the author's name should be before the comma.
Answer
Corrected (lines 5 -6)
Remark 3
The abstract lacks a description of the main research results.
Answer
Lines 30-32 deleted: “The discovered patterns of protein and metabolite dynamics revealed rearrangement of energy metabolism in favor of anaerobic pathways that most likely indicated the plant response to the hypoxic stress”.
New lines 33-36 added: “Thus, rearrangement of energy metabolism was obvious on the 10th day after inoculation (d.a.i.). Specifically, redirection of the energy metabolism from the oxidative to the anaerobic pathway was observed. This might be a part of the plant adaptation response to tumor-induced hypoxic stress, which most likely included activation of sugar signaling”.
Remark 4
The references cited in the preface do not include those from 2022-2024, do not mention metabolomics related studies, and do not mention the correlation analysis between proteomics and metabolomics. The result analysis also does not include the correlation analysis between proteomics and metabolomics.
Answer
We agree with the reviewer – new references are needed. Therefore, 12 new references from 2022-24 were included.
Remark 5
The statement 'On the 10th and 21st days after inference' mentioned in line 141 contradicts the 10th and 22nd days in the results analysis.
Answer
The statement was deleted as relating to another part of the study not included in the article.
Remark 6
The RNA isolation mentioned in line 143 did not yield any relevant results in the result analysis.
Answer
Phrase “On the 10th and 21st days after inoculation, corresponding hypocotyl fragments and growing crown galls were harvested, frozen in liquid nitrogen and stored at -80° C before protein and RNA isolation” were deleted as relating to another part of the study not included in the article.
Remark 7
The header of Table 3 is quite strange, pay attention to where some bold words are located.
Answer
Сorrected (line 587)
Remark 8
“rpm” should be converted to “g”.
Answer
Line 161, line 166, line 170 - Since this is not a centrifuge but a shaker, recalculation to g is not possible
Remark 9
Part of the fonts in Figure 3 are blurry or even ghosted, and Figure 5 is not clear enough.
Answer
Figures 3 и 5 were re-done - clearer markings have been made to make the information easier to understand (lines 396 - 401 and 645 – 650)
Remark 10
The line numbers of lines 677 and 678 are not on the right side, blocking some of the content.
Answer
Corrected in the word file – see lines 683 - 684
Remark 11
Some abbreviations that appear for the first time do not have their corresponding full names, such as OD and CV (is CV ≤ 60 correct?) FC, RI, and RSD. In addition, 'GDM' should be 'GMD'.
Answer
CV ≤ 60 is correct
Line 137 added “optical density” (OD)
Line 208 added “coefficient of variation”
Line 216 added “fold changes”
Line 251 added “retention indices”
Line 282 added “relative standard deviation”
Line 248 corrected GDM to GMD
Remark 12
In lines 135 and 162, there is no space between the numbers "28oC" and "4oC" and their units, while in line 159, there is a space between the numbers "37 oC" and their units. Similarly, please check for similar issues.
Answer
Line 132 and in all similar cases - checked and corrected
Remark 13
Whether the numbering, first line suspension, and blank space before and after paragraphs of the secondary headings under different primary headings are inconsistent, please check the formatting of the entire text.
Answer
Checked, redone
Remark 14
Please check if "2792" in line 299 is correct.
Answer
It is correct
Remark 15
What is the basis for dividing proteins into 5 clusters in Table 1? Can the width of Table 1 be further widened to make it more aesthetically pleasing?
Answer
Clusters were defined according to the occurrence of individual proteins among differentially expressed in paired comparisons. The detailed explanation can be found in the legend below Table 1. The corresponding changes are done: “which were formed according to the occurrence of individual proteins among differentially expressed in paired comparisons – lines 567-569 (see also explanation/legend for Table 1 – lines 589 - 591)”
Remark 16
In the discussion, relevant model of protein and metabolite changes should be summarized in figure to reflect the rearrangement of energy metabolism that is favor of anaerobic pathways expressed in the abstract.
Answer
We agree with the reviewer and provide Figure 8 addressing this aspect. Discussion is modified accordingly and the figure is referred in the text (line 846).
Remark 17
The subheading on line 746 is neither numbered nor formatted differently from the subheading on line 971, and is 4 pages long. It is recommended to divide it into several subheadings.
Answer
Done as suggested – now we have four sub-chapters here – see lines 791, 835, 974 and 1058.
Remark 18
On line 1001, the reference number is missing in square brackets.
Answer
Line 1088 two references are provided
Remark 19
For other questions, please refer to the annotations in the PDF.
Answer
We are very grateful to the reviewer for all the identified typos and errors in writing, we have tried to carefully correct everything according to the remarks.

Reviewer 3 Report
Comments and Suggestions for Authors
The topic of this study is very interesting. The manuscript provides useful data regarding the molecular mechanisms behind the response to inoculation with Agrobacterium tumefaciens. The authors did a thorough job in terms of methodology to achieve the results. The results and discussion are very comprehensive. The English needs to be revised and improved.

The main issue with the article is the English, which makes some parts confusing. The English needs to be revised and improved.
Author Response
We thank the Reviewer for the thoughtful review and highly appreciate the valuable comments and suggestions to improve the manuscript. Following these advices we performed all required changes in corresponding sections, as indicated in the following rebuttal addressing each aspect. These changes are referred with the line numbers which are also highlighted as comments in the text
Reviewer 3
Remarks
General remark
… the main issue with the article is the English, which makes some parts confusing.
The English needs to be revised and improved
Answer
English is revised and corrected throughout the whole text. Please see the improved version submitted
Remark 1
L44 – In the sentence “The strategy of the agrobacterial partner relies on integration of its T-DNA (a part of the virulent plasmid) into the plant genome”, I suggest “The strategy of the agrobacterial partner relies on the integration of its T-DNA (a portion of the virulent plasmid) into the plant genome.
Answer
We thank the reviewer for the suggestion. The sentence is changed accordingly (lines 43-45)
Remark 2
L47 – Eliminate the word 'transfer' because T-DNA implicitly includes that concept.
Answer
We thank the reviewer for the suggestion. The sentence is changed accordingly (lines 45 – 46)
Remark 3
L48 – I suggest replacing 'in plant genome' with 'into the plant genome'.
Answer
We thank the reviewer for the suggestion. The sentence is changed accordingly (lines 47-48)
Remark 4
L49 - In this sentence, '...encoding the enzymes of auxin, cytokinin...', the authors may have intended to say '...encoding enzymes involved in auxin, cytokinin...'
Answer
We thank the reviewer for the suggestion. The sentence is changed accordingly (lines 48 -49)
Remark 5
L51 - I suggest replacing 'featured' with ' characterized'.
Answer
We thank the reviewer for the suggestion. The sentence is changed accordingly in a slightly another way (lines 50 – 51)
Remark 6
L53 – ‘…of ethylene, abscisic acid and jasmonates’.
Answer
We thank the reviewer for the suggestion. The sentence is changed accordingly (lines 51 – 52)
Remark 7
L53-55 – ‘…It results in the enhanced proliferation of plant parenchyma cells and the formation of the crown gall tumor [8], which is accompanied by increased expression of the genes normally involved in meristems development [9].’
Answer
We thank the reviewer for the suggestion. The sentence is changed accordingly in a slightly another way (lines 52 – 55)
Remark 8
L56-58 – ‘In this context, the interaction of plants with A. tumefaciens provides a convenient model to study the alterations in plant metabolism and meristems function that accompany plant development.’
Answer
We thank the reviewer for the suggestion. The sentence is changed accordingly (lines 56 – 58)
Remark 9
L58-60 – ‘Indeed, as tumor formation is underlied by the loss of systemic control over cell proliferation, a comprehensive study of these aspects could provide insights into the mechanisms behind this control.’
Answer
We thank the reviewer for the suggestion. The sentence is changed accordingly (lines 58 – 60)
Remark 10
L60-63 – ‘This knowledge is of principal importance, as the molecular mechanisms underlying tumor formation in plants are fundamentally different from those known in animals and occur much less frequently in nature.’
Answer
We thank the reviewer for the suggestion. The sentence is changed accordingly (lines 60 – 63)
Remark 11
L65 - The authors should replace the expression 'model object.' In my opinion, 'object' is not appropriate in this context.
Answer
We thank the reviewer for the suggestion. We replaced it with “model plant” (line 65)
Remark 12
L67-70 – ‘Post-genomic research methods (commonly referred to as “omics technics”) represent a powerful toolbox in plant developmental biology and have proven useful for studying A. tumefaciens-induced tumors in radish..’
Answer
We thank the reviewer for the suggestion. The sentence is adjusted to requirements of English grammar (lines 68 – 70)
Remark 13
L70-72 – ‘Thus, the proteomics data may provide insights into the functional relationshipes between changes in the genome and transcriptome and the characteristic morphological and biochemical alterations in phenotype.’
Answer
We thank the reviewer for the suggestion. The sentence is changed accordingly (lines 70 – 72)
Remark 14
L73-76 – ‘Recently, a combination of transcriptomics and proteomics approaches was employed to uncover the fine molecular mechanisms underlying growth and development in radish (Raphanus sativus L.). To achieve this, the transcriptome of the growing radish taproot was analyzed by RNA-seq in parallel to a comprehensive proteomics survey conducted using iTRAQ.’
Answer
We thank the reviewer for the suggestion. The sentence is adjusted to requirements of English grammar (lines 73 – 77)
Remark 15
L81-84 – I suggest that the authors rewrite this part of the text. In terms of English, it is not the most correct.
Answer
The part is re-written as following:
“Despite the impressive analytical power of the modern proteomics, this methodology was only minimally applied to the study of the A. tumefaciens-induced tumors so far [17–19]. Indeed, highly-efficient liquid chromatography (LC)-based shotgun proteomics is still to be employed for the study of plant responses to A. tumefaciens” (lines 82 – 85)
Remark 16
The objectives are clear; however, I suggest that the authors explain what is meant by the term 'late proteome'. In the objectives, I further suggest that the authors include the metabolome analysis, as it was conducted.
Answer
The part is corrected as following:
“Therefore, here we report, to the best of our knowledge, the first comprehensive study of radish proteome dynamics in response to agrobacterial infection with a focus on late (i.e. those occurring after the first week post-inoculation) alterations in proteome associated with tumor development and accompanying metabolic and physiological changes.” (lines 86 – 90)
Remark 17
Plant growing and Inoculation of radish seedlings with Agrobacterium tumefaciens:
In this section, the authors should explain how many seeds were planted and how many were inoculated with Agrobacterium. In this section, the sample nomenclature should also be included.
Answer
All requested changes are done in the Plant growing and Inoculation section – please, see three sites marked in the text (lines 123 – 124, 129, 138 - 139).
Remark 18
L133 - Please italicize Agrobacterium tumefaciens.
Answer
We understand the idea of the reviewer but we cannot deitalize the species name in this case, as the title itself is italic (line 135).
Remark 19
Protein isolation and tryptic digestion:
The authors should briefly explain the total protein extraction method, specifying the amount of plant material used in the extraction.
Why were the protein concentration results validated by SDS-PAGE? What is the importance of performing SDS-PAGE in this situation?
Answer
The requested changes are done: we provide the sample weights and describe the protein isolation procedure (which is published and well-known) in Supplementary information. SDS-PAGE was employed as a orthogonal method for cross-validation. See the corresponding changes in the text (lines 149, 150, 167)
Remark 20
I also suggest placing the statistical analysis in a separate section.
Answer
We thank the reviewer for the suggestion. However, we don’t believe that we need such a section, as we use proteomics- and metabolomics- specific standard methods which are also incorporated in softwares. Therefore, we describe them in corresponding sections in the text, but not arrange a specific section to combine them. We believe that it is counterproductive.
Remark 21
Figure 1: In the images, I suggest adding arrows to indicate the crown galls, including this information in the image caption as well.
Answer
All requested corrections are done: arrows are provided and the legend is complemented by corresponding extension (line 273 – 274)
Remark 22
Table S1-5 - In the table caption, the sample nomenclature should be indicated.
Answer
The issue is corrected
Remark 23
Protein isolation and tryptic digestion: Given the extent of the results, in my opinion, this section is not essential. Much of the information presented here could be described in the Materials and Methods, and it is not necessary to have a section to describe that some samples were eliminated for being considered outliers.
Answer
We thank the Reviewer for this remark. However, we would like to keep this section to fit better to the world proteomics standards and to ensure transparency of our data and protocols
Remark 24
Figure 2 - In the figure caption (L309), what does n=3 correspond to?
Answer
N=3 corresponds to the number of independently prepared digests (in this case – replicate number) - line 313
Remark 25
Figure 3 - In the figure caption, I suggest changing "left" and "right" to "A" and "B", respectively, placing "A" and "B" in the figure. This makes interpretation easier.
Answer
We thank the Reviewer for this remark. However, it is not two panels, but one panel with two sides. Therefore, we think that making “A” and “B” would confuse the reader (line 396)
Remark 26
L396 - Remove "after inoculation" since "d.a.i." already implies this.
Answer
Corrected accordingly (line 402)
Remark 27
L481 - Please, remove the repeated word.
Answer
Corrected accordingly (line 487)
Remark 28
L566-572 - In this paragraph, I am confused with the nomenclature CLAVA-567 TA3/ESR-RELATED 22, etc….
Answer
We are not sure that we are understand the issue. The nomenclature used in this paragraph is correct and state of the art. But for us the principal thing is - this nomenclature from Uniprot (line 570)
Remark 29
Analysis of the primary metabolome - Was the metabolome analysis targeted or untargeted? This should always be clearly explained, including in the Materials and Methods.
Answer
We provided the information in Material and methods. But detailed description is given in the reference Shumilina et al, which is provided (lines 240 – 241, 608, 627).
Remark 30
Figure 6 and 7 - Figures A and C are too small. I suggest increasing the size of the figures by reorganizing Figures 6 and 7
Answer
The figures are corrected as required (line 658, 750).
Remark 31
In my opinion, the discussion is very comprehensive, encompassing and justifying the results obtained. However, the English should be improved. The way it is written makes some paragraphs confusing.
Answer
The discussion part is re-written, and English is improved
Remark 32
L789-791- In this part of the text, I understand that the authors are describing the results obtained; however, they include a reference, which gives the impression that these results are from another study and not their own. I suggest rewriting this part or removing the reference at the end of the sentence.
Answer
We thank the reviewer for this remark – corrected appropriately (line 857)
Remark 33
L851-862 - This paragraph is confusing. I suggest rewriting it.
Answer
We agree with the reviewer – in that form this paragraph was a bit difficult for the reader. Now we complemented it with a summarising sentence and explanatory graphical summary (Figure 8). We hope, with all this, the paragraph is not confusing anymore (lines 846, 947, 949).
Remark 34
L864 - In this sentence, I would remove the word 'Obviously'.
Answer
We agree with the reviewer and remove it (line 950)
Remark 35
L905-906 - The parenthesis is missing a closing bracket.
Answer
Corrected accordingly (line 994)
Remark 36
L949 - Please, remove the repeated word.
Answer
Corrected accordingly (line 1036)
Remark 37
L981 - Please italicize Agrobacterium.
Answer
Corrected accordingly (line 1068)

Round 2
Reviewer 2 Report
Comments and Suggestions for Authors
Although the manuscript has made significant improvements, there are still the following issues:
1. In practical applications, the coefficient of variation is usually much less than 100% (or 1, if expressed in decimal form). When the coefficient of variation approaches or exceeds 100%, it indicates that the data has a high degree of discreteness, and the differences between the data are very large. You can imagine that in a bar chart, if CV=1, the length of the error line expressed in standard deviation (SD) will be equal to the height of the bar (Mean). So, I think CV≤60 should be changed to 'CV≤60%'. Please confirm this issue.
2. The description in Table 1 is not clear enough. The "log2fold change (FC)" in the header can be changed to "log2FCs" (there should be a blank column on the left to separate it from the three columns on the left). The annotations under the four categories "10d, 22d, MI, AI" are different from the previous annotations and can easily confuse readers. It is recommended to renumber them, such as "R-10d, R-22d, R-MI, R-AI". Otherwise, readers may not understand which two ratios were used to classify the previous 5 clusters.
3. It is recommended to remove the diagonal arrows and equal signs from Table 1, including the corresponding content indicated below.
4. Use the multiplication sign "×" instead of the letter "x" (especially in italics) in the p and q values in Table 1.
5. Is the change in the ratio of the p and q values in parentheses in Table 1 the same as the change in the cluster they belong to? If they are different, it is recommended to remove them.
6. The font of "No" in Table 1 is different from other fonts. In addition, each row of the same cluster should be aligned at the top, so that the lines added below the 3rd and 4th clusters can be removed; Otherwise, a horizontal line should be added between any two clusters.
Comments on the Quality of English Language
The language of the manuscript deserves further improvement.
Author Response
We thank the Reviewer for a thoughtful review and appreciate the valuable comments and suggestions to improve the manuscript. Following this advice, we have made all necessary changes to the relevant sections as indicated in the following rebuttal addressing each aspect. We also highlighted these changes in the text as comments.
Reviewer 2
Remark 1
In practical applications, the coefficient of variation is usually much less than 100% (or 1, if expressed in decimal form). When the coefficient of variation approaches or exceeds 100%, it indicates that the data has a high degree of discreteness, and the differences between the data are very large. You can imagine that in a bar chart, if CV=1, the length of the error line expressed in standard deviation (SD) will be equal to the height of the bar (Mean). So, I think CV≤60 should be changed to 'CV≤60%'. Please confirm this issue.
Answer
Line 204: We thank the reviewer for this important specification, as the coefficient of variation is indeed calculated as a percentage. Thus, changed accordingly.
Remark 2
The description in Table 1 is not clear enough. The "log2fold change (FC)" in the header can be changed to "log2FCs" (there should be a blank column on the left to separate it from the three columns on the left). The annotations under the four categories "10d, 22d, MI, AI" are different from the previous annotations and can easily confuse readers. It is recommended to renumber them, such as "R-10d, R-22d, R-MI, R-AI". Otherwise, readers may not understand which two ratios were used to classify the previous 5 clusters.
Answer
Line 547: The table has been corrected accordingly to the reviewer's recommendations, in particular, column boundaries have been added to make it easier to read. Abstract headers have been renamed to “R-10d, R-22d, R-MI, R-AI”. ‘log2fold change (FC)’ in the header was changed to “log2FCs”.
Remark 3
It is recommended to remove the diagonal arrows and equal signs from Table 1, including the corresponding content indicated below.
Answer
We understand the point of the reviewer, but we woud like to keep this presentation form as it facilitates understanding and these sings are essential for describing differential expression processes in clusters
Remark 4
Use the multiplication sign "×" instead of the letter "x" (especially in italics) in the p and q values in Table 1.
Answer
All corrections have been made
Remark 5
Is the change in the ratio of the p and q values in parentheses in Table 1 the same as the change in the cluster they belong to? If they are different, it is recommended to remove them.
Answer
These are not ratios, but the values for each group, italics indicate MI and AI groups, respectively. Presentation of p and q is absolutely mandatory for proteomics works. Because of this, these values cannot be removed from the table.
Remark 6
The font of "No" in Table 1 is different from other fonts. In addition, each row of the same cluster should be aligned at the top, so that the lines added below the 3rd and 4th clusters can be removed; Otherwise, a horizontal line should be added between any two clusters.
Answer
All corrections have been made
